# Beyond Forecasting: Compositional Time Series Reasoning for End-to-End Task Execution

## Abstract

In recent decades, there have been substantial advances in time series models and benchmarks across various individual tasks, such as time series forecasting, classification, and anomaly detection. Meanwhile, compositional reasoning in time series prevalent in real-world applications (e.g., decision-making and compositional question answering) is in great demand. Unlike simple tasks that primarily focus on predictive accuracy, compositional reasoning emphasizes the synthesis of diverse information from both time series data and various domain knowledge, making it distinct and extremely more challenging. In this paper, we introduce Compositional Time Series Reasoning, a new task of handling intricate multistep reasoning tasks from time series data. Specifically, this new task focuses on various question instances requiring structural and compositional reasoning abilities on time series data, such as decision-making and compositional question answering. As an initial attempt to tackle this novel task, we developed TS-Reasoner, a program-aided approach that utilizes large language model (LLM) to decompose a complex task into steps of programs that leverage existing time series models and numerical subroutines. Through a comprehensive set of experiments, we demonstrate that our simple but effective TS-Reasoner outperforms existing standalone reasoning approaches. These promising results indicate potential opportunities in the new task of time series reasoning and highlight the need for further research.

## 1 Introduction

Over the past few decades, research in time series analysis has heavily focused on improving the performance of individual tasks such as time series forecasting, anomaly detection, and time series classification (De Gooijer & Hyndman, 2006; Kirchgässner et al., 2012; Zong et al., 2018; Dau et al., 2019; Hamilton, 2020; Jin et al., 2024). These results have benefited various areas such as risk assessment in finance, disease diagnose in healthcare, pandemic modeling in public health and event detection in natural and social science (Tsay, 2005; Cao et al., 2022; 2023c; Kamra et al., 2021; Team & Murray, 2020; Penfold & Zhang, 2013; Cheng et al., 2021; Sharma et al., 2021; Zhang et al., 2021).

However, most real-world applications demand multi-step reasoning, where well-established tasks should serve as intermediate steps. A typical example is forecasting future energy supply (Zheng et al., 2022), in which scientists must integrate domain knowledge with statistical analysis. The process begins with the examination of time series data to forecast future signals with statistical methods, following by formulating constraints based on domain expertise to refine predictions. Another typical example is analyzing climate time series (Mudelsee, 2010), where it is not only necessary to predict what happens next but also crucial for experts to comprehend the fundamental physical laws governing such data. For instance, a climate scientist studying the impact of greenhouse gas emissions on global temperature might employ advanced time series analysis for multi-step reasoning, including *forecasting* future temperature trends and variability, *analyzing cross-correlations* between different climate indicator variables (e.g., CO2 levels, ocean temperatures, ice cover), *detecting anomalies* in weather patterns, *simulating* various emission scenarios, and *imputing missing* historical climate data. Traditionally, these various analyses would be involved with many different specialists - climatologists for temperature forecasting, oceanographers for sea-level analysis, atmospheric scientists for greenhouse gas modeling, and data scientists for anomaly detection. Such manual solutions to scientific projects are labor-intensive, and often requiring long time ranging

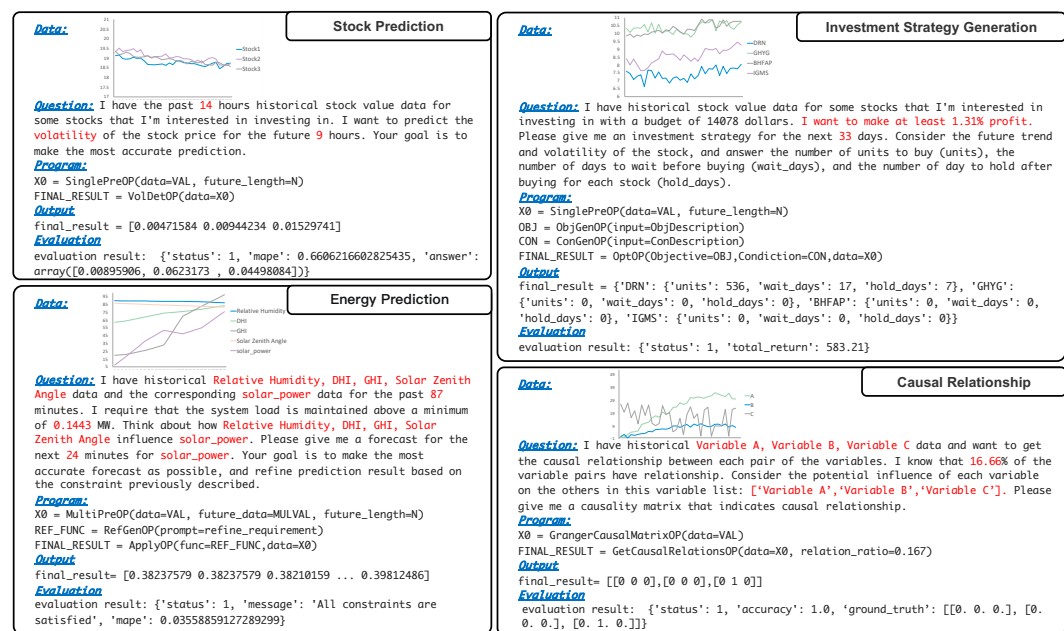

Figure 1: Examples of End-to-End Tasks on Time Series. These tasks require the model to perform multi-step reasoning on time series data.

from months to years. Additionally, the intermediate tasks like forecasting are typically optimized independently, leading to inefficiencies. In contrast, an end-to-end task framework for time series reasoning, particularly with advancements in large language models (LLMs), can optimize these subtasks cohesively. This integrated paradigm can reduce time and resource requirements, resulting in more accurate and timely predictions.

Nevertheless, this aim of end-to-end time series task execution is challenging due to the lack of exploration at the intersection of structural reasoning and numerical computation of temporal signals. State-of-the-art time series foundation models (Ansari et al., 2024; Woo et al., 2024; Cao et al., 2023b) excel in handling temporal patterns but lack reasoning abilities over context observed in Large Language Models (LLMs) (Edwards et al., 2024). This dichotomy is particularly evident in compositional time series tasks requiring structural multi-step reasoning, since current time series models (Wu et al., 2021; 2022), despite their sophistication, are primarily designed for single-task inference with predefined task definitions. This paradigm limits existing time series models' applicability to diverse, compositional problems. This gap in time series analysis opens up significant research opportunities in the realm of multi-step reasoning, where models are required to synthesize multiple pieces of information over time series.

With the rise of LLMs, complex logic-based reasoning on data with various modality has become readily feasible (Qiao et al., 2022; Huang & Chang, 2022), providing a promising opportunity for the study of complex time series analysis and compositional reasoning. Existing work has explored reasoning across various domains, such as chain-of-thought (CoT) reasoning in natural language (Wei et al., 2022), VisProg in computer vision (Gupta & Kembhavi, 2023), and Thought-of-Table for tabular data (Wang et al., 2024). However, research focused specifically on complex and compositional time series reasoning remains limited. Most studies involving large language models and time-series modality still primarily emphasize individual tasks (e.g., forecasting) (Jin et al., 2024), feature understanding(Fons et al., 2024) and one-step question answering (Merrill et al., 2024). To the best of our knowledge, multi-step and compositional reasoning in time series remains very under-explored.

In this paper, we make an initial attempt at bridging the aforementioned gap by proposing a new paradigm of Complex and Compositional Time Series Reasoning. It shifts the focus from standard predictive tasks to those requiring sophisticated reasoning processes. More specific examples are shown in Fig. 1. Furthermore, we develop a program-aided reasoning approach, which utilizes LLMs' in-context learning ability to decompose complex tasks into structured programs for multi-

step reasoning. This flexible approach named as TS-Reasoner[1] is different from traditional program-aided reasoning systems (Gupta & Kembhavi, 2023) in that it supports the creation of custom modules and adapts to external knowledge and/or users-specified constraints. The goal of TS-Reasoner is to empower domain experts by providing a tailored solution that reduces the labor-intensive nature of multi-step analysis tasks. Unlike general-purpose models such as LLMs, which aim to address a broad range of tasks but often lack precision and perform poorly on domain-specific challenges, TS-Reasoneris designed to integrate domain expertise and support specialized modules. This targeted approach ensures that the model is not only versatile but also highly effective in delivering results in specialized fields. For evaluation, we compiled new datasets in finance and energy domains and constructed a series of commonly-asked and most-concerned questions (D'Amico et al., 2022; Lee et al., 2019; Zidan & El-Saadany, 2013) that requires complex reasoning and compositions. Through extensive experiments in two application domains, we demonstrate that TS-Reasoner consistently achieves better results than state-of-art reasoning approaches in domain-specific evaluations. These promising results reveal potential opportunities in complex and compositional time series reasoning and underscore the importance of further exploration into this new task.

## 2   RELEVANT WORKS

**Time Series Analysis Tasks and Models**    Classical time series analysis encompasses several key tasks that leverage patterns and trends in data over time. These tasks include forecasting, imputation, classification, and anomaly detection. Each of these tasks serves unique purposes across various application domains, highlighting the significance of time series analysis.

To tackle time series analysis, researchers have made significant contribution over the years. Early methods were mainly task-specific, where each individual task was addressed by a dedicated model optimized for its specific purpose, resulting in a fragmented approach. Recently, inspired by the emergence of large language models, there has been a shift toward general-purpose Large Time Series Models. Notable contributions include work by (Gruver et al., 2024), who simply encoded time series as strings, and (Jin et al., 2023), who converted time series into language representations through alignment. (Cao et al., 2023b) and (Pan et al., 2024) incorporated decomposition techniques and prompt design, enabling generalization to unseen data and multimodal scenarios. (Zhou et al., 2023) adapted GPT-2 as a general-purpose time series analysis model, extending it to various tasks. Additionally, (Talukder et al., 2024) utilized VQVAE as a tokenizer for transformers, while (Ansari et al., 2024) employed scaling and quantization techniques for embedding time series. These models are designed to handle multiple preset tasks and are jointly pre-trained on diverse datasets. However they still operate under predefined task definitions, which limits their ability to perform complex and compositional reasoning. As a result, while they can handle multiple tasks, they lack the flexibility to adapt to more intricate scenarios that require a deeper understanding on task instructions and composition of different tasks or concepts.

**Complex and Compositional Reasoning with Pre-trained Foundation Models**    Large Language Models (LLMs) have demonstrated significant capabilities in managing complex reasoning tasks by emulating human cognitive processes, especially when incorporated with appropriate in-context samples (Huang & Chang, 2022; Qiao et al., 2022; Ahn et al., 2024; Qu et al., 2024). The Chain of Thought (CoT) prompting method (Wei et al., 2022) is a prime example, encouraging models to articulate intermediate reasoning steps (i.e. *rationales*) before reaching a conclusion. This method improves performance in multi-step logical deductions by transparently demonstrating the thought process leading to an answer as well as expanding the expressive power of Transformer architecture Feng et al. (2023); Merrill & Sabharwal (2023). Following CoT, more rationale dependency structures are proposed to capture more reasoning paradigms, such as Tree-of-Thoughts, Graph-of-Thoughts and Self-Consistent CoT Yao et al. (2023a;b); Wang et al. (2022).

Moreover, program-based reasoning (Zhu et al., 2022; Jung et al., 2022; Zhou et al., 2022; Khot et al., 2022; Creswell & Shanahan, 2022; Gao et al., 2023) represents an advanced form of eliciting complex reasoning from LLMs. This involves framing reasoning tasks as code generation, where the model is trained or prompted to understand and manipulate logical constructs akin to programming

---

[1]Demo video explaining how TS-Reasoner works can be found at `https://www.youtube.com/watch?v=FCB7atczbfc&t=1s`

(Shi et al., 2023). Such approach enables LLMs to make use of heterogeneous modules to tackle various data modalities. For example, VisProg (Gupta & Kembhavi, 2023) allows LLM to call vision modules for visual reasoning tasks, and Chain-of-Table (Wang et al., 2024) enables LLM to acquire better performance on tabular data analysis. The tabular data reasoning is analogous to time series (Dong et al., 2019; He et al., 2024; Hu et al., 2024; Du et al., 2021). However, tabular data is typically static and presented as discrete records, while time series emphasizes dynamic changes and focuses on trends and patterns that change over time, requiring consideration of temporal dependencies and continuity. Consequently, the datasets related to tabular data reasoning are mostly concerned with tasks such as information retrieval (He et al., 2024), information completion (Bandyopadhyay et al., 2019), and fact verification (Chen et al., 2019; Zhang et al., 2020), often overlooking the dynamic characteristics of the time dimension. As a result, existing methods for tabular data are not directly suitable for address complex time series questions, especially when temporal dynamics play a key role in such questions.

Recent works also explored the performance of LLM on time series understanding and question answering. Fons et al. introduces a comprehensive taxonomy of time series features and utilizes a synthetic dataset to evaluate LLM performance on tasks such as feature detection, classification, and arithmetic reasoning. In contrast, Merrill et al. focuses on more difficult question-answering tasks involving time series data, including challenges such as etiological inference, which require simultaneous understanding over natural language and time series inputs. However, both studies are constrained by their focus on individual tasks, lacking exploration of multi-step or compositional reasoning, thus limiting their applicability to more advanced inferential processes.

## 3 TASK DEFINITION

In this section, we first define compositional time series reasoning, which involves the synthesis of information from temporal data in conjunction with task-specific instructions and contextual knowledge.

**Definition 3.1** (**Compositional Time Series Reasoning**). *Let* $\mathbf{x}$ *denote a time series, which is a sequence of data points indexed in time order. Let* $\mathbf{C}$ *represent the context, which encompasses the* ***task instruction*** *and additional external information. The primary objective of time series reasoning is to derive a set of rationales* $\mathbf{R} = (r_1, r_2, \ldots, r_n)$ *that are conditional on the inputs* $\mathbf{x}$ *and* $\mathbf{C}$ *in an step-by-step manner. Each* $r_i$ *addresses a single sub-task related to the time series, e.g.* $r_1$ *tackle missing value imputation,* $r_2$ *tackle forecasting, and* $r_3$ *tackle numerical reasoning and optimization. Then we can generate the final answer* $\mathbf{y}$ *to the task based on both rationales and the input, mathematically expressed as:*

$$r_i = f(r_1, ..., r_{i-1}, \mathbf{x}, \mathbf{C}) \quad \Rightarrow \quad \mathbf{y} = g(\mathbf{R}, \mathbf{x}, \mathbf{C}) = g(r_1, ..., r_n, \mathbf{x}, \mathbf{C})$$

Here, $f$ is a function that constructs the next rationale bsed on previous rationales and inputs; $g$ is a function that maps the generated rationales $\mathbf{R}$ along with the inputs $\mathbf{x}$ and $\mathbf{C}$ to the final response $\mathbf{y}$, and $g$. The rationales $\mathbf{R}$ serve as intermediary results or conclusions that facilitate the reasoning process and may exhibit various probabilistic structural dependencies. The most common one is the sequential dependency, on which we define **Chain of Thought (CoT)**(Feng et al., 2023):

$$r_i \sim p_\theta(r_i|r_1, r_2, ..., r_{i-1}, \mathbf{x}, \mathbf{C}) \tag{1}$$

where $p_\theta$ is a Large Language Model (LLM), and every rationale is generated fully by the LLM based on auto-regressive decoding.

In this paper, we apply an alternative paradigm due to the nature of time series as structured data, which is **Program-based Reasoning**, whose mathematical expression is as follows:

$$f_i \sim p_\theta(f_1, f_2, ..., f_{i-1}, \mathbf{x}, \mathbf{C}) \quad \mathbf{y} = g(\mathbf{R}, \mathbf{x}, \mathbf{C}) = g(f_1, ..., f_n, \mathbf{x}, \mathbf{C}) \tag{2}$$

where $p_\theta$ is a LLM that can generate code, and $f_1, ..., f_i$ are program sentences generated by $p_\theta$.

## 4 DATASET AND TASKS

Our dataset[2] is primarily built for three major categories of tasks: decision making on financial data, compositional question answering about finance market and energy usage, and causal mining on

---

[2]We will release the data and materials for public use.

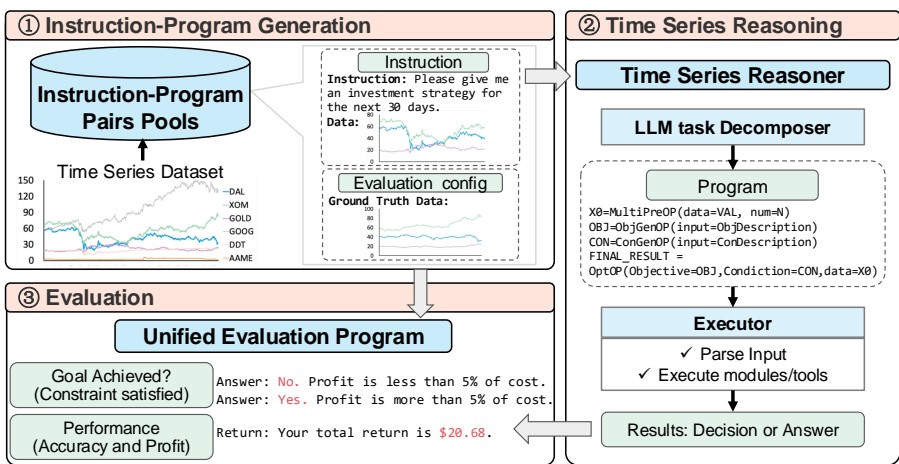

Figure 2: The proposed pipeline for evaluation and time series reasoner. Top left: Instruction, Data, Evaluation Config sampled Instruction-Program Generator. Bottom Left: Evaluation given model output and evaluation config. Right:TS-Reasoner performs task decomposition and program execution to obtain final answer.

synthetic dataset (Denis et al., 2003; Gonzalez-Vidal et al., 2019; Cao et al., 2023a). Among these tasks, decision making present unique challenges for evaluation, as they cannot be easily assessed through simple comparisons with ground truth answers like traditional question answering tasks. To address this, we innovatively proposed an instruction/program pool to abstract the evaluation process for these tasks, shown in Fig. 2. Specifically, we designed a set of appropriate instructions for each application domain, in which each instruction is paired with an evaluation configuration that outlines the criteria for success. Given a response, our unified evaluation program (shown in Algorithm 1) determines whether the answer is successful based on the evaluation configuration, reporting both the success rate and end task specific performance metrics. For instance, in the context of financial decision making, we assess whether the given decision is compliant with the given budget and evaluate the corresponding outcomes such as the total profit. This comprehensive approach allows us to effectively evaluate performance across diverse tasks that extend beyond conventional time series analysis, providing a clear picture of both success rates and overall effectiveness.

```python
def Evaluator(response, ground_truth_data, eval_config):
    #obtain relevant context from question
    context = eval_config['context']
    #obtain needed constraints specified in question
    constraint = eval_config['constraint']
    #constraint verification
    flag = check_constraint(response=response,context=context, constraint=constraint)
    #task specific evaluation
    performance = task_specific_eval(eval_config["task_name"],data=ground_truth_data)
    return flag, *performance
```

Algorithm 1: The Unified Evaluation Program.

## 4.1 DECISION MAKING

In our decision-making task, we focus on investment portfolio decisions within the financial market, which requires the ability to synthesize information from multiple areas such as trend recognition, risk assessment, and numerical optimization based on human expertise (Bonaparte et al., 2014). For each test sample, historical stock prices of interest are provided alongside immediate future data. The historical data includes natural language questions articulating investment goals—such as maximizing profit—as well as constraints, including budget limitations, expected profit ratios, and acceptable loss ratios. The question is generated according to the following template:

**Question Template (Decision Making on Financial Time Series)**   *I have historical stock value data for some stocks that I'm interested in investing in with a budget of {budget} dollars. [1. I want to make at least {profit percent}% profit. 2. I have a risk tolerance of {risk percent}%. 3. I want to allocate no more than {allocation budget} dollars to {stock name}.] Please give me an investment strategy for the next {future length} {data resolution: day or hour}s. {output requirement}.*

## 4.2 COMPOSITIONAL QUESTION ANSWERING

In our compositional question answering task, we primarily focus on financial markets and load-related issues in the energy sector. Specifically, each test sample provides the model with a natural language question and relevant time series historical data, such as stock prices and energy supply data. The questions are generated by the following templates:

**Question Template (Compositional Question Answering on Energy Supply Time Series)**   *I have historical {influence variables} data and the corresponding target variable data for the past {historical length} minutes. [1. I need to ensure that the maximum allowable system load does not exceed {load value} MW. 2. I require that the system load is maintained above a minimum of {load value} MW. 3. I must monitor the load ramp rate to ensure it does not exceed {constraint value} MW for each time step. 4. I need to manage the load variability so that it does not exceed {constraint value} MW over the given period.] Think about how {influence variables} influence {target variable}. Please give me a forecast for the next {future length} minutes for target variable. Your goal is to make the most accurate forecast as possible, refine prediction result based on the constraint previously described, {output requirement}.*

**Question Template (Compositional Question Answering on Financial Time Series )**   *I have the past {historical length} hours historical stock value data for some stocks that I'm interested in investing in. [1. I want to predict the volatility of the stock price 2. I want to predict the stock price ] for the {future length} hours. Your goal is to make the most accurate prediction. Please give me your prediction, {output requirement}.*

## 4.3 CAUSAL MINING

For causal mining, we synthesize a set of data grounded in domain knowledge related to climate science, finance, and economics. Specifically, we generate a series of multivariate time series data based on established causal relationships and meteorological principles. Each test sample consists of a time series dataset of various variables, accompanied by a natural language instruction that asks the model to uncover the causal dependencies between the given time series. For details on data generation process, please refer to section C.4. The reasoning model must infer dependencies based on the given data and instructions. The evaluation framework then measures the model's performance by comparing its inferred causal relationships with the ground truth. The questions are generated by the following template:

**Question Template (Causal Analysis)**   *I have historical {variable names} data and want to get the causal relationship between each pair of the variables. I know that {ratio}% of the variable pairs have relationship. Consider the potential influence of each variable on the others in this variable list: variable names. {output requirement}.*

## 5 PROGRAM-BASED TIME SERIES REASONING

When handling time series data, methods that rely solely on large language models (LLMs) for reasoning, such as the Chain-of-Thought (CoT) approach, often struggle with understanding numerical information (Zhang et al., 2024). These models, while powerful in generating logical inferences, are prone to making errors in calculations or failing to adhere to numerical constraints that are crucial in tasks involving time series analysis. These shortcomings underscore the necessity for programmatic assistance in reasoning processes. To mitigate such errors, we propose a framework that

supplements LLM-based reasoning with program-based decomposition and leverages the in-context learning ability of LLMs.

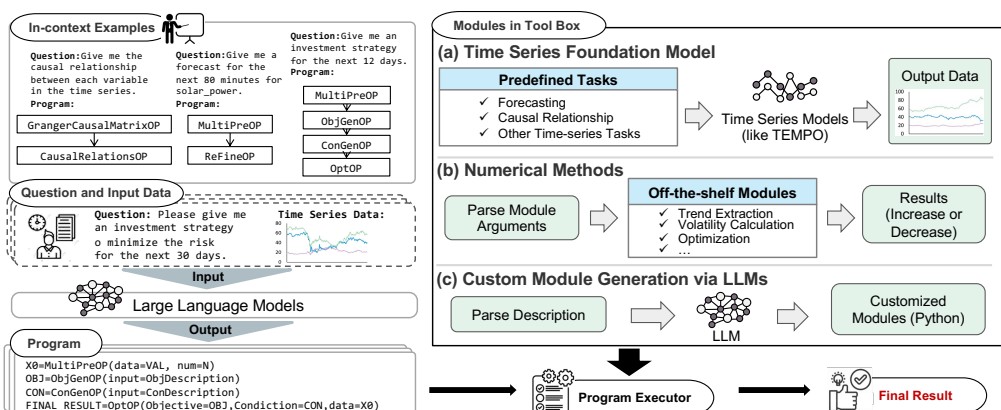

Figure 3: The pipeline of TS-Reasoner. The LLM work as task decomposer, which learn from in-context examples to decompose task instances as programs. Then a program executor will call modules in our tool box to run relevant programs in the given order to obtain final result.

**Task Decomposer** The proposed framework handles time series data by integrating program-based decomposition and task-specific models. As illustrated in Fig.3, the core idea revolves around a programmatic task decomposition engine, which we refer to as the "Problem Decomposer". This component is responsible for disassembling complex tasks into a series of smaller, manageable subtasks, each described in a programmatic manner. These subtasks are subsequently addressed by distinct processing modules, enabling the framework to provide robust, step-wise solutions to time series-related problems. In TS-Reasoner , We use ChatGPT-3.5-turbo as our task decomposer. We leverage the in-context learning ability of pretrained language model and construct question-program pairs as in-context examples. The in context examples are carefully constructed so that the samples questions are equally distributed across the four question types (Financial Investment Strategy, Future Stock Characteristic Prediction, Energy Load Perdiction with known knowledge, Causal Relation). As shown in Fig. 3 , every in-context sample is a question program pair where the question is described in natural language and the program is pseudo-code like. Please refer to section C.1 for prompts given to task decomposer.

The decomposition of tasks allows for targeted processing through three types of modules, each specialized for different aspects of the reasoning process:

**Time Series Model Modules:** These modules are grounded in foundation time series models and are primarily responsible for handling standard operations such as forecasting, anomaly detection, trend analysis, and other predictive or diagnostic tasks. Their purpose is to leverage established models in the field to process data-driven subtasks with high precision and efficiency.

**Numerical Method Modules:** A second class of modules focuses on numerical and statistical methods. These modules are particularly adept at performing quantitative manipulations on the data, such as extracting trends, computing ranges, and conducting basic arithmetic or statistical analyses. The application of numerical techniques allows for a clearer interpretation of time series dynamics, particularly in tasks where precise quantitative reasoning is required.

**Custom Module Generation via Large Language Models (LLMs):** The third type of module addresses a significant challenge in time series reasoning: the handling of external knowledge and user-specific instructions that cannot be predefined. In many real-world scenarios, users may incorporate unique external knowledge (i.e. maximum or minimum of the value range in forecasting) or requirement, which are often expressed in natural language within the input instructions. To handle such custom requirements, the framework includes a "Custom Module Generation Function," which calls on a large language model (LLM) to interpret the natural language directives. The LLM translates these personalized constraints and objectives into programmatic code, wrapping them into a callable module that integrates seamlessly with the broader reasoning process.

| Task Requirement | TS-Reasoner | | | CoT + code | | | CoT | | |
|---|---|---|---|---|---|---|---|---|---|
| | SR(%) | AAP | RAP | SR(%) | AAP | RAP | SR(%) | AAP | RAP |
| Profit Percent | **59.2** | **243.31** | **32.34** | 10.0 | 38.75 | -172.21 | 18.0 | 0.0 | -210.97 |
| Risk Tolerance | **96.0** | **54.54** | **-46.04** | 24.0 | 4.36 | -96.22 | 10.0 | 0.0 | -100.58 |
| Budget Allocation | **90.0** | **37.12** | **7.57** | 32.0 | -98.54 | -128.09 | 6.0 | -0.41 | -29.96 |

Table 1: The success rate and performance of TS-Reasoner against other baselines on desicion making. SR stands for Success Rate; AAP stands for Absolute Average Profit. RAP is the Relative Average Profit compared to vanilla strategy. In Profit Percent and Budget Allocation task, we aim at improving the profit. Thus positive RAP is expected. In Risk Tolerance, the model is required to first ensure the risk and minimize the profit reduction. A negative RAP indicates a more conservative model in terms of risk management compared to vanilla strategy. **Bold** indicates the best results.

Together, these three module types execute the sequence of tasks generated by the Problem Decomposer. Once the subtasks are processed, the system produces corresponding outputs and traces, ensuring transparency and traceability in the solution path. For details on available modules in the toolbox, please refer to section C.2.

# 6 EXPERIMENTS

In this section, we conducted a series of comparative experiments to assess the performance of various models across our defined tasks of decision making, compositional question answering, and multi-domain causal mining. Our baseline models included the Chain of Thought (CoT) approach and CoT + code approach. For the most competitive result, we used ChatGPT-4-turbo. In CoT prompting, we outline the steps for the model to think about and directly return result. In CoT + code setting, we provide CoT prompts that outline steps to take and additionally allows the model the generate code that we execute to obtain result. Fore more details on the prompts, please refer to section C.3. Through these experiments, we aimed to measure not only the performance of the outputs but also the models' ability to adhere to constraints and optimize outcomes within the complex frameworks of financial markets and energy usage. By systematically analyzing the strengths and weaknesses of each approach, we seek to elucidate the most effective strategies for leveraging large language models in practical decision-making, compositional question answering and causal inference tasks.

## 6.1 DECISION MAKING

**Evaluation Protocol** In decision making, the overall objective and specific user requirements may be different. For this reason, we respectively report the performance of models on each kind of instances. The user's main objective is to maximize the total profit/ minimize the loss. The customized requirements can be generally divided to: Profit Percent Guarantee (the decision needs to guarantee the minimum profit percent that the user expected), Risk Tolerance (the volatility of the investment portfolio must be within an expected range), and Budget Allocation (control the budget for a specific stock). In evaluation, we focus on two types of metrics: success rate (SR), absolute average profit (AAP) and relative average profit (RAP). The strict success rate is defined as the percentage of test samples that did not violate any constraint and requirements. The average absolute profit is the profit that the model made on all successful instances. The relative average profit is defined as the relative profit gain over the vanilla investment strategy that do not consider the requirements in the instructions. In Profit Percent and Budget Allocation task, we aim at improving the profit over the vanilla strategy. In Risk Tolerance, the model is required to first ensure the risk and minimize the profit reduction over the vanilla strategy.

**Overall Performance** Table 1 shows the performance of TS-Reasoner and baseline reasoning approaches. It is evident that TS-Reasoner generally outperforms CoT and CoT + code in terms of strict success rate, particularly in financial decision-making. TS-Reasoner achieves high success rates in risk tolerance and budget allocation, and performs moderately in profit percent, which is intuitive as guaranteeing profit is always harder than constraining loss and budget. Also, it notable that TS-Reasoner has a higher relative profit gain over vanilla strategy and behaves more conservatively in risk control scenarios (although still maintaining overall positive absolute profit). On the

| Task | Reasoning Steps | TS-Reasoner | | CoT + code | | CoT | |
|---|---|---|---|---|---|---|---|
| | | SR(%) | MAPE(std) | SR(%) | MAPE(std) | SR(%) | MAPE(std) |
| Stock Future Price Prediction | 1 | **100.0** | **0.042(0.030)** | 98.00 | 0.043(0.031) | **100.0** | 0.058(0.047) |
| Stock Future Volatility Prediction | 2 | **100.0** | **0.748(0.691)** | 90.00 | 0.848(0.181) | 88.00 | 0.865(0.158) |
| Energy Power w/ Max Load | 3 | **97.87** | **0.101(0.339)** | 85.10 | 0.226(0.402) | 53.20 | 0.120(0.230) |
| Energy Power w/ Min Load | 3 | **97.83** | **0.084(0.104)** | 73.91 | 0.682(1.337) | 54.30 | 0.279(0.500) |
| Load Ramp Rate in Energy Power | 3 | **100.0** | **0.060(0.153)** | 89.58 | 0.167(0.281) | 75.00 | 0.062(0.170) |
| Load Variability Limit in Energy Power | 3 | 93.88 | 0.288 (0.385) | 85.71 | 0.265(0.472) | 55.10 | **0.243(0.391)** |

Table 2: The overall success rate and performance of our model against other baselines on compositional QA. SR stands for Success Rate; MAPE is the Mean Absolute Percentage Error. **Bold** indicates the best results.

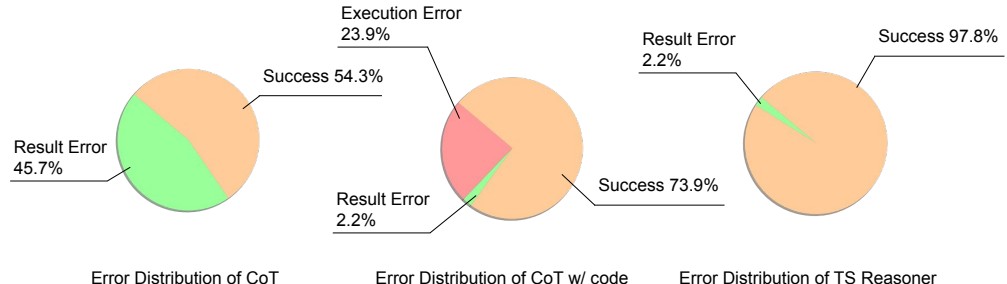

Figure 4: Error distribution of different approaches on QA of Energy Power w/ Min Load.

task of Profit Percent and Budget Allocation, TS-Reasoner consistently improve the profit over the vanilla strategy in the instruction. On the task Risk Tolerance, we can see that TS-Reasoner acquired the minimum loss when ensuring highest success rate. In contrast, the baselines completely lose competitiveness against vanilla strategy.

## 6.2 COMPOSITIONAL QUESTION ANSWERING

**Evaluation Protocol**   In compositional QA, the various questions may lead to different reasoning steps. For this reason, we respectively report the performance of models on each kind of instances. Specifically, we applied the data from finance (stock price) and energy power supply. For finance, we mainly tackle the prediction on price, volatility, which are relatively simple. For energy supply, we consider the energy power supply forecast with external requirement attached such as the max and min load regularization for the system or load ramp rate and variability limit. Given such requirements, TS-Reasoner needs to additionally refine the results based on the specified external knowledge. In evaluation, we focus on two types of metrics: success rate (SR) and Mean Absolute Percentage Error (MAPE). The success rate is defined as the percentage of test samples in which the model successfully execute the tasks and did not violate any constraint in the instructions (e.g. Energy Power load constraint/Requirement).

**Overall Performance**   Table 2 shows the performance of TS-Reasoner and baseline reasoning approaches. It is evident that TS-Reasoner generally outperforms CoT and CoT + code in terms of both success rate (SR) and MAPE. We can observe that as reasoning steps increase, TS-Reasoner shows a clear advantage over CoT and CoT + code. For simpler tasks with 1-2 steps, the performance across all models is relatively similar. However, as tasks become more complex, TS-Reasoner consistently outperforms both baselines, with significantly higher success rates (SR) and better MAPE.

**Error Analysis**   In Fig. 4, we present a case study on the energy load prediction task when minimum load is specified. The analysis examines how the error rate varies in different approaches. By introducing program-aided reasoning, result errors are substantially alleviated, which are usually bottlenecked by numerical errors from LLM. Meanwhile, although CoT w/ code compress the rate of result error, it additionally introduce execution error, which implies problematic code generated by LLM. In contrast, TS-Reasoneris able to eliminate execution errors due to robustly tested modules. This result provides more insight in the advantage of program-based reasoning in time series

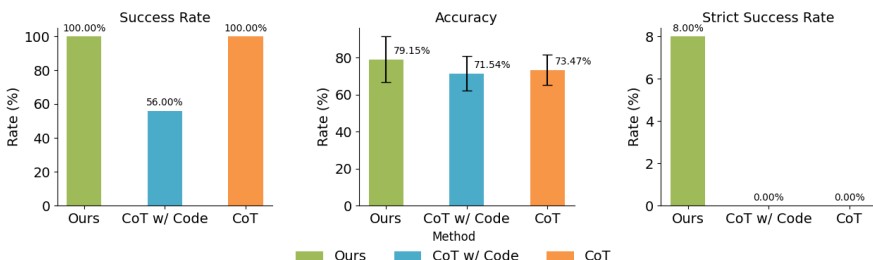

Figure 5: The overall success rate and performance of TS-Reasoner against other baselines on Causal Relationship Recognition.

reasoning. The errors include the neglection of critical constraints and requirement, mismanagement of numerical computations, problematic integration of intermediate output. For concrete examples of error cases, please refer to Appendix B

### 6.3 CAUSAL RELATIONSHIP RECOGNITION

**Evaluation Protocol**   In causal relationship recognition, TS-Reasoner is given multi-variable time series, a description to the data, and expert knowledge of percentage of true relationships. The model need to incorporate the expert knowledge and the causal discovery tools based on directed acyclic graph to infer the probable causal relationship across multiple variables. In evaluation, we focus on three types of metrics: success rate (SR), causal relationship accuracy (CRA) and causal graph accuracy (CGA). The success rate is defined as the percentage of test samples in which the model successfully execute the task and did not violate any constraint in the instructions (e.g. the percentage of the causal relationships). The causal relationship accuracy is defined as the accuracy of classifying each pair of variable as causally related or not. The causal graph accuracy is defined as the percentage of test samples of which all causal relationships are correctly classified.

**Overall Performance**   In the causal relation recognition task, TS-Reasoner outperform the CoT and CoT + code on all metrics, as shown in Fig. 5. It is also noticeable that the performance of all methods on CGA, which is the hardest evaluation metric, are not satisfactory. Specifically, both CoT-based methods acquired 0.0 accuracy, which means that for any given test instance, none of these approaches can correctly infer all pairs causal relationships within it. Although TS-Reasoner slightly outperforms the baselines, the result is still very modest, opening opportunities for future works on addressing challenges under this setting.

## 7 CONCLUSION

In this work, we introduced the task of Complex and Compositional Time Series Reasoning. To support this task, we curated a specialized dataset from multiple domains and designed a domain-specific evaluation protocol for rigorous assessments. Building on this foundation, we developed TS-Reasoner, a simple yet effective model that integrates program-based task decomposition with LLMs and domain-specific modules. By combining time series models, numerical techniques, and LLM-generated custom modules, TS-Reasonerachieves a balance between numerical precision and flexibility, enabling it to handle a variety of tasks requiring domain expertise and personalized decision-making. Unlike purely LLM-driven methods that often suffer from numerical inaccuracies and limited domain adaptability, TS-Reasoner leverages a structured decomposition approach to mitigate these limitations. This integration provides a versatile framework for domain experts to streamline complex multi-step analyses. For future work, we plan to expand the dataset to encompass more diverse domains, explore techniques for incorporating world knowledge from LLMs into the reasoning process, and enhance the toolbox with more powerful modules. These advancements aim to further elevate TS-Reasoner's capabilities in compositional and domain-specific time series analysis.

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

## A    ANALYSIS FOR DIFFERENT IN-CONTEXT SAMPLES.

In Fig. 6, we analyze how the in-context samples impact the correctness of the program generated by the model. Specifically, we report how the percentage of the program that are correct varies with the number of in-context samples. As we can see, as the number increases, the correctness of generated program improves. When 16 samples are provided, the model is able to always generate correct programs.

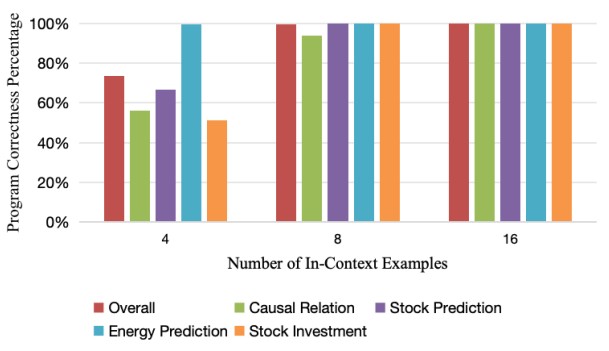

Figure 6: Program Correctness Rate under varying numbers of in-context samples.

## B    ERROR EXAMPLES

In this section, we present the example errors (shown in Fig. 7 and 8) from TS-Reasoner, CoT, and CoT + code approaches.

| Error Type | Constraint Violation | Result Error (Shape Misalignment) | Constraint Violation |
|---|---|---|---|
| User Instruction | I have historical stock value data for some stocks that I'm interested in investing in with a budget of 4587 dollars. I have a risk tolerance of 7.7%. Please give me an investment strategy for the next 36 days. Consider the future trend and volatility of the stock, give me the rationale and then answer with a formatted python dictionary with each stock name as keys and include the number of units to buy (units), the number of days to wait before buying (wait_days), and the number of day to hold after buying for each stock (hold_days). | I have historical Dew Point, Relative Humidity, Solar Zenith Angle data and the corresponding load_power data for the past 108 minutes. I require that the system load is maintained above a minimum of 0.70 MW. Think about how Dew Point, Relative Humidity, Solar Zenith Angle influence load_power. Please give me a forecast for the next 68 minutes for load_power. | I have historical Wind Speed, Relative Humidity data and the corresponding wind_power data for the past 167 minutes. I need to manage the load variability so that it does not exceed 0.0305 MW over the given period. Think about how Wind Speed, Relative Humidity influence wind_power. Please give me a forecast for the next 65 minutes for wind_power. Your goal is to make the most accurate forecast as possible, refine prediction result based on the constraint previously described. |
| Output | Final_value: {'ENVX': {'units': 120, 'wait_days': 5, 'hold_days': 30}, 'VTC': {'units': 15, 'wait_days': 10, 'hold_days': 40}, 'MSN': {'units': 3000, 'wait_days': 3, 'hold_days': 20,…} | Final_value: [0.87 0.89 ... 0.88] Final_value.shape = (66,) | Final_value: [ 0. ... 0.03612244 ...] |
| Evaluation Result | {'status':0, 'message':'Cost of investment exceeds budget', 'failed_value':11464.72} | {'status':0, 'message': 'Prediction Shape Misalignment, Expecting output of length 68', 'error':1} | {'status': 0, 'message': 'Predicted load variability exceeds the maximum allowable limit of 0.0305 MW. |

Figure 7: Examples of Result Errors in TS-Reasoner and CoT approach.

## C    PROMPT AND TOOL

### C.1    PROMPT FOR TS-REASONER

Return only programs, using the specified operation functions. Do not return any results like dictionaries or lists. You must accurately learn the relationship between the question and the required operations. You must choose correct operations for each question. Do not use other irrelevant operations.

Question:

I have historical Variable A, Variable B, Variable C, Variable D, Variable E, Variable F data and want to get the causal relationship between each pair of the variables. I know that 70.0% of the

| Error Type | Execution Error | Execution Error |
|---|---|---|
| User Instruction | I have historical stock value data for some stocks that I'm interested in investing in with a budget of 4587 dollars. I have a risk tolerance of 7.7%. Please give me an investment strategy for the next 36 days. Consider the future trend and volatility of the stock, give me the rationale and then answer with a formatted python dictionary with each stock name as keys and include the number of units to buy (units), the number of days to wait before buying (wait_days), and the number of day to hold after buying for each stock (hold_days). Exclude any comment in the python code. | I have historical stock value data for some stocks that I'm interested in investing in with a budget of 4913.151824981799 dollars. I have a risk tolerance of 1.65%. Please give me an investment strategy for the next 22 hours. Consider the future trend and volatility of the stock, give me the rationale and then answer with a formatted python dictionary with each stock name as keys and include the number of units to buy (units), the number of hours to wait before buying (wait_hours), and the number of hour to hold after buying for each stock (hold_hours). |
| Execution | An error occurred: name 'budget' is not defined. | An error occurred: can't multiply sequence by non-int of type 'float' final value. |
| Evaluation Result | {'status':0, 'message':"received NoneType"} | {'status':0, 'message':"received NoneType"} |

Figure 8: Examples of Execution Errors occurred CoT + code approach.

variable pairs have relationship. Consider the potential influence of each variable on the others in this variable list: ['Variable A', 'Variable B', 'Variable C', 'Variable D', 'Variable E', 'Variable F']. Please provide 2d numpy matrix with binary values to indicate whether each pair of variables has a relationship.

Program:

X0 = GrangerCausalMatrixOP(data=VAL)

FINAL_RESULT = GetCausalRelationsOP(data=X0, relation_ratio=RATIO)

Question:

I have the past 65 days historical stock value data for some stocks that I'm interested in investing in. I want to predict the volatility of the stock price for the future 18 days. Your goal is to make the most accurate prediction .Please give me your prediction, return a 1d numpy array with the predicted volatility of each stock.

Program:

X0=SinglePreOP(data=VAL, future_length=N)

FINAL_RESULT = VolDetOP(data=X0)

...(more in-context samples)

Question:

I have historical Relative Humidity, Temperature data and the corresponding wind_power data for the past 191 minutes. I need to ensure that the maximum allowable system load does not exceed 1.33 MW. Think about how Relative Humidity, Temperature influence wind_power. Please give me a forecast for the next 56 minutes for wind_power. Your goal is to make the most accurate forecast as possible, refine prediction result based on the constraint previously described, and return the result as a 1D array.

Program:

Follow previous examples and answer my last question in the same format as previous examples within markdown format in "'python"'. Only include output steps in the python markdown, do not repeat my question or include 'Program:' in python markdown. Do not use irrelevant operations, only use operations that are necessary for the question.

## C.2 Available Modules in Toolbox

The tasks defined in this paper, along with their corresponding input-output formats and the tools utilized, are summarized in Table 3 below.

Specifically, the detailed description of the Predefined Tools is as follows:

| Task | Input-Output | Tools |
|------|-------------|-------|
| Future Stock Prediction | Input: Time Series + Question
Output: Time Series | SinglePreOP, MultiPreOP, TrendPreOP, RefGenOP |
| Stock Investment | Input: Time Series + Instruction
Output: Investment Strategy | MultiPreOP, ApplyOP, ObjGenOP, ConGenOP, OptOP |
| Energy Prediction | Input: Time Series + Question
Output: Time Series | MultiPreOP, ApplyOP, RefGenOP |
| Causal Relation | Input: Time Series + Instruction
Output: Causal Relationship | GrangerCausalMatrixOP, GetCausalRelationsOP |

Table 3: Overall performance of causal relationship recognition.

- SinglePreOP: This operator is primarily used for univariate time series forecasting, leveraging advanced time series large language models to accurately predict future sequences. The input consists of two variables: 'data', representing the input historical time series, and 'future_length', specifying the number of future steps to predict. The output is a time series of length 'future_length'.

- VolDetOP: This operator performs volatility detection on time series by utilizing a volatility detection algorithm. The input consists of a single variable, 'data', which represents the time series to be analyzed. The output is the volatility result for the input time series, which can be classified into one of three categories: 'volatility clustering', 'persistent volatility', or 'non-volatility'.

- TrendPreOP: This operator performs trend detection on time series by utilizing a trend detection algorithm. The input consists of a single variable, 'data', which represents the time series to be analyzed. The output is the trend result for the input time series, classified into one of three categories: 'increasing', 'decreasing', or 'steady'.

- MultiPreOP: This is a multivariate time series forecasting operator used to accurately predict the target variable based on multiple covariates. The input consists of three variables: 'data', 'future_data', and 'future_length'. 'data' represents the historical time series of several covariates and one target variable, 'future_data' provides the future time series of several covariates, and 'future_length' specifies the forecast length for the target variable. The output is the predicted time series of the target variable with a length of 'future_length'.

- RefGenOP: This operator is primarily used to generate a corresponding function based on the requirement described in a time series forecasting task. The generated function can then be applied to 'data' to meet the specified conditions. The input consists of a single variable, 'requirement', which represents the requirement description from the task. The output is a function generated according to the task requirement.

- ApplyOP: This operator functions as an executor, applying the input function to the corresponding data. The input consists of two variables: 'func', which represents a function (such as a requirement function generated by RefGenOP), and 'data', which represents the data to be processed. The output is obtained by passing 'data' through the 'func' function.

- ObjGenOP: This operator is designed to generate an optimization objective for an optimal investment strategy based on the user's investment goals. The input consists of a single variable, 'obj_information', which contains the user's description of their investment objectives (including expected profit, etc.). The operator utilizes large language models to interpret the investment-related information provided by the user and generates an optimization objective function. The output is the optimization objective function derived from the input investment information.

- ConGenOP: This operator is designed to generate a constraint function based on the user's investment-related constraints. The input consists of a single variable, 'con_information', which contains the user's description of their investment constraints (such as budget limits and acceptable risk tolerance). The operator leverages large language models to interpret the investment-related information provided by the user and generates a corresponding constraint function. The output is the constraint function derived from the input investment information.

- OptOP: This operator is primarily used for generating the optimal stock investment strategy. The input consists of four variables: 'data', 'constraint', 'future_length', and 'objective'. 'data' repre-

sents the historical stock data to be invested in, 'constraint' denotes the investment constraint function (such as budget constraints, return rate constraints, etc.), 'future_length' specifies the number of future investment time steps, and 'objective' represents the investment objective function. This operator leverages large language models to understand stock data, investment goals, and constraints to automatically generate the optimal investment strategy. The output is the optimal investment strategy based on all input information.

- GrangerCausalMatrixOP: This operator is used to calculate the Granger causality relationship between each pair of variables in a time series dataset. The input consists of a single variable, 'data', which is composed of the historical data of C time series. The output is a matrix of shape [C, C], where each element represents the significance (p-value) of the causal relationship between the corresponding variables.

- GetCausalRelationsOP: This operator is used to determine significant causal relationships based on a Granger causality matrix, which stores the p-values of the causal relationships between variables. A threshold parameter, 'relation_ratio', is applied to identify significant causal relationships, and the output is a binary matrix indicating which variables have significant causal links. The input consists of two variables: 'data', which is the precomputed causality matrix, and 'relation_ratio', which is the threshold. The output is a matrix composed of zeros and ones, reflecting the causal relationships.

## C.3 CoT Prompt

### (1) Stock Future Price Prediction CoT

You are an experienced data scientist specializing in time series forecasting. I will provide you with a list of multiple time series. You must generate only predictions for the following questions, returning only the predictions without any codes, markdown formatting or extra characters. question

Chain of Thought: Step 1: Understand and parse the input data from text. Must clarify the number of all stocks. Step 2: Choose an appropriate model: You will select an appropriate time series forecasting model (e.g., ARIMA, LSTM, etc.) based on the input data. Make sure the model is suitable for forecasting the next n steps. Step 3: Apply the model: You will apply the chosen model to each time series (each column in the input data) and generate predictions for the next n steps. Step 4: Return the prediction results: Output the future predictions directly as 'predictions=List([List(),List(),...,])'.

Requirement: Do not return any codes, just the final results 'predictions=List([List(),List(),...,])'. Please ensure that the output number of stocks is correct and the predicted length is accurate. Simply output the future predictions as a list. The predictions should be stored in a variable called 'predictions' and output the list directly.

### (2) Stock Future Price Prediction CoT with Code

You are an expert in time series forecasting. You need to perform a forecasting tasks. Generate only Python code, no markdown or extra characters, for the task below: question

Instructions: 1. Input is a 2D numpy array 'data' of shape [L, C], where L is the historical data length and C is the number of time series. 'n' is the future length to predict. 2. Define a function that predicts n steps for each time series using models like ARIMA or LSTM. Ensure the model outputs all n steps of predictions. 3. Store your output in the variable called 'predictions'. 'predictions' should be a nested list: 'predictions = [[step1, step2, ..., stepn] for each time series]', Your prediction should be a 2d array of shape [n,C].

Requirements: - Define data = np.array([]) as placeholders and do not include any hardcoded data in the data variable. - Ensure the code is fully executable and 'n' is set from the prompt.

### (3) Stock Future Volatility Prediction CoT

You are an experienced data scientist specializing in time series forecasting. I will provide you with a list of multiple time series. You must generate only predictions for the following questions, returning only the predictions without any codes, markdown formatting or extra characters. question

Chain of Thought: Step 1: Understand and parse the input data from text. Must clarify the number of all stocks. Step 2: Choose an appropriate model: You will select an appropriate time series forecasting model (e.g., ARIMA, LSTM, etc.) based on the input data. Make sure the model is

suitable for forecasting the next n steps. Step 3: Apply the model: You will apply the chosen model to each time series (each column in the input data) and generate predictions for the next n steps. Step 4. Calculate the volatility of each time forcasted time series. Volatility refers to the standard deviation of a time series. Step 5: Return the prediction results: Output the future predictions directly as 'predictions=List([volatility1, volatility2,...])'.

Requirement: Do not return any codes, just the final results 'predictions=List([volatility1, volatility2,...])'. Please ensure that the output number of stocks is correct and the predicted length is accurate. Simply output the future predictions as a list of length equals to stocks. The predictions should be stored in a variable called 'predictions' and output the list directly.

### (4) Stock Future Volatility Prediction CoT with Code

You are an expert in time series forecasting. You need to perform a forecasting tasks. Generate only Python code, no markdown or extra characters, for the task below: question

Instructions: 1. Input is a 2D numpy array 'data' of shape [L, C], where L is the historical data length and C is the number of time series. 'n' is the future length to predict. 2. Define a function that predicts n steps for each time series using models like ARIMA or LSTM. Ensure the model outputs all n steps of predictions. 3. Calculate the volatility of each time forcasted time series 4. Store your output in the variable called 'predictions'. 'predictions' should be a list od length C: 'predictions = [volatility1, volatility2,...]'.

Requirements: - Define data = np.array([]) as placeholders and do not include any hardcoded data in the data variable. - Ensure the code is fully executable and 'n' is set from the prompt.

### (5) Stock Future Trend Classification CoT

You are an experienced data scientist specializing in time series forecasting. I will provide you with a list of multiple time series. You must generate only predictions for the following questions, returning only the predictions without any codes, markdown formatting or extra characters. question

Chain of Thought: Step 1: Understand and parse the input data from text. Must clarify the number of all stocks. Step 2: Choose an appropriate model: You will select an appropriate time series forecasting model (e.g., ARIMA, LSTM, etc.) based on the input data. Make sure the model is suitable for forecasting the next n steps. Step 3: Apply the model: You will apply the chosen model to each time series (each column in the input data) and generate predictions for the next n steps. Step 4. Calculate the trend of each time forcasted time series. Trend refers to increasing, decreasing and unknown. Step 5: Return the prediction results: Output the future predictions directly as 'predictions=List([trend1, trend2,...])'.

Requirement: Do not return any codes, just the final results 'predictions=List([trend1, trend2,...])'. Please ensure that the output number of stocks is correct and the predicted length is accurate. Simply output the future predictions as a list of length equals to stocks. The predictions should be stored in a variable called 'predictions' and output the list directly.

### (6) Stock Future Trend Classification CoT with Code

You are an expert in time series forecasting. You need to perform a forecasting tasks. Generate only Python code, no markdown or extra characters, for the task below: question

Instructions: 1. Input is a 2D numpy array 'data' of shape [L, C], where L is the historical data length and C is the number of time series. 'n' is the future length to predict. 2. Define a function that predicts n steps for each time series using models like ARIMA or LSTM. Ensure the model outputs all n steps of predictions. 3. Extract the trend of each time series forcasted time series (increasing, decreasing, unknown) 4. Store your output in the variable called 'predictions'. 'predictions' should be a list od length C: 'predictions = [trend1, trend2,...]'.

Requirements: - Define data = np.array([]) as placeholders and do not include any hardcoded data in the data variable. - Ensure the code is fully executable and 'n' is set from the prompt.

### (7) Electricity Prediction CoT

You are an expert in time series forecasting. You need to perform a forecasting task on electricity data. Generate only the final results, no markdown, code, or extra characters, for the task below:

question. Given future covariate data $future\_data$ and the prediction length $future\_length$, please predict the values of the target variable.

Chain of Thought: 1: Understand and parse the input data from text. Here are several historical time series, where only one time series is the target variable and the other time series are covariates. 2. Select a multiple regression model to model the relationship between covariates and the target variable, such as the LinearRegression model. 3: Based on the future covariate data, use the model from step two to predict the target variable of the length $future\_length$. 4: Refine your predictions if necessary to meet the required constraints and store the final results in predictions. 5. If the predicted length exceeds $future\_length$, please truncate it and then return the predictions of length $future\_length$.

Requirements: The predictions must be returned as a completely list predictions of length $future\_length$. Ensure that the output is directly the result without any code, markdown, or extra characters. Only return the list predictions with length $future\_length$.

**(8) Electricity Prediction CoT with Code**

You are an expert in time series forecasting. You need to perform a forecasting task on electricity data. Generate only Python code, no markdown or extra characters, for the task below: question

Instructions: 1. Input is a 2D numpy array 'data' of shape [L+n, C], where L is the historical data length and C is the number of time series. 'n' is the future length to predict. The first C-1 columns are covariates, and the last column is the target variable. 2. Define a function that predicts n steps for the target variable given the covariates of the future n steps. 3. Store your output in the variable called 'predictions'. 'predictions' should be a list of length n: 'predictions = [step1, step2,...]'. 3. Given the constraint, refine your prediction and store the refined prediction in the variable called 'predictions'.

Requirements: - Define data = np.array([]) as placeholders and do not include any hardcoded data in the data variable. - Ensure the code is fully executable and 'n','L' is set from the prompt. - Do not include any comments, especially when you call the function.

**(9) Causal Relation CoT**

You are an expert in time series causality analysis. You need to perform a causality analysis task. Generate only predictions, no any codes, no any markdown or extra characters, for the task below: question

Instructions: 1. Create a 2D numpy array 'data' of shape [hist_len, stock_len], where hist_len is the data length and stock_len is the number of time series. 2. Implement a function that analyzes the causal relationship between the time series using Granger causality test. 3. Store your output in the variable called 'causal_relations'. 'causal_relations' should be a 2d array of shape [stock_len, stock_len] 4. Given the constraint, refine your causal relations and store the refined causal relations in the variable called 'causal_relations'. 5. Your final result should contain binary values (0 or 1) indicating the presence or absence of causality between the time series. 6. Store the final output in the variable called 'predictions'. 7. You don't need to consider self-causality. The ratio mentioned in the constraint is the ratio of the number of causal relations to the total number of possible causal relations excluding self-causality.

Requirements: - Ensure that the final output in the predictions variable strictly follows a two-dimensional array format, containing only binary values (0 or 1) that signify the presence or absence of causality between the time series. - Must return only the predictions with a shape [stock_len, stock_len].

**(10) Causal Relation CoT with Code**

You are an expert in time series causality analysis. You need to perform a causality analysis task. Generate only Python code, no markdown or extra characters, for the task below: question

Instructions: 1. Input is a 2D numpy array 'data' of shape [L, C], where L is the data length and C is the number of time series. 2. Define a function that analyzes the causal relationship between the time series using Granger causality test. 3. Store your output in the variable called 'causal_relations'. 'causal_relations' should be a 2d array of shape [C, C] 4. Given the constraint, refine your causal relations and store the refined causal relations in the variable called 'causal_relations'. 5. Your final

result should contain binary values (0 or 1) indicating the presence or absence of causality between the time series. 6. Store the final output in the variable called 'predictions'. 7. You don't need to consider self-causality. The ratio mentioned in the constraint is the ratio of the number of causal relations to the total number of possible causal relations excluding self-causality.

Requirements: - Define data = np.array([]) as placeholders and do not include any hardcoded data in the data variable. - Ensure the code is fully executable. - Do not include any comments, especially when you call the function. - For any package import, import the functions directly.

**(11) Stock Investment CoT**

You are an expert in stock investment. You need to perform a stock investment task. Generate only final strategy, no any codes, no any markdown or extra characters, for the task below: question

Instructions:

1. Create a pandas DataFrame data with columns stock_columns, representing historical stock prices over hist_len hours. 2. Implement a prediction function for forecasting future stock prices using an appropriate model. 3. Develop an objective function to optimize the total expected profit, considering the budget and the other constraints. 4. Apply constraints to ensure that units are non-negative, both wait_resolutions and hold_resolutions are at least zero, and their sum does not exceed future_len. 5. Calculate the optimal investment strategy, optimizing the expected profit while adhering to the constraints. 6. Format the optimized strategy into a dictionary named predictions that details the investment strategy for each stock.

Output Requirements: 1. Directly return the optimized strategy as a dictionary formatted as follows: "investment target" (name of the stock): "units": number of shares to buy and should be no smaller than 0 "wait_resolutions": number of resolutions to wait before buying the stock and should be greater than or equal to 0 "hold_resolutions": number of resolutions to hold the stock for and should be strictly less than the future length n , 2. Ensure the strategy is feasible within the provided constraints and budget. 3. Exclude any extraneous comments or code annotations in the output.

**(12) Stock Investment CoT with Code**

You are an expert in stock investment. You need to perform a stock investment task. Generate only Python code, no markdown or extra characters, for the task below: question

Instructions: 1. Input is a pandas dataframe 'data' of shape [L, C], where L is the historical data length and C is the number of time series. The column names are the stock names, and the values are the stock prices. 2. Define a function that predicts the future price of each stock using a suitable model. 3. Design an objective and constraint function to optimize the investment strategy. 4. Optimize the investment strategy based on the functions you designed. Store the optimized strategy in the variable called 'predictions'. 5. The output format should be "investment target" (name of the stock): "units": number of shares to buy and should be no smaller than 0 "wait_resolutions": number of resolutions to wait before buying the stock and should be greater than or equal to 0 "hold_resolutions": number of resolutions to hold the stock for and should be strictly less than the future length n ,

Requirements: - Define data =pd.DataFrame() as placeholders and do not include any hardcoded data in the data variable. - Ensure the code is fully executable. - Do not include any comments, especially when you call the function. - For any package import, import the functions directly.

C.4   CASUAL MINING DATA GENERATION

Now you are a Time series data scientist, please help me to write the code to generate some synthetic data in real world Time series domain, you should save the data into "*/data.csv":

Now suggesting you should construct a series data based on a relation matrix and the correlation ratio for different influence factor, you should notice the following points,for time step I want you to generate 500 time steps:

1. data correlation: the multi variable should be correlated, sample: which A first influence B, then B have influence on C or D, there should be some time delay, as the influence on other staff needs time.

2. data trend: there should be some trend in the data, like the data is increasing or decreasing.

3. data: seasonality there should be some seasonality in the data, like the data is periodic.

4. data noise: the noise should be added to the data, as the real world data is not perfect.

5. data background: the data should have some real world background, you should first think about different real world data, and provide a description for the variable and time series data, then generate the data using the code. CoT Sample: Q: Approximate Relation Ratio: 0.5 Relation Matrix:

|   | $A$ | $B$ | $C$ | $D$ |
|---|---|---|---|---|
| $A$ | 1 | 1 | 0 | 0.5 |
| $B$ | 0 | 1 | 0 | 1 |
| $C$ | 0 | 1 | 1 | 1 |
| $D$ | 0 | 0 | 0 | 1 |

- A influences B and D, and itself.
- B influences D, and itself.
- C influences B and D, and itself.
- D influences only itself.

variable size: 4 A: Scenario: Sales Data of a Chain of Stores Over Time Let's assume we are generating synthetic data,the variable size for the data is 4. for the daily sales of multiple stores across a chain, the sales numbers are influenced by:

1. Advertising (A): The level of advertising spend directly impacts the sales of each store. After a delay, this starts influencing sales. 2. Sales (B): The sales numbers for each store are influenced by both the advertising and local seasonal events. 3. Economic Factors (C): Broader economic trends, like GDP growth or unemployment rates, also impact sales. These factors show a delayed and more subtle influence over time. 4. Customer Sentiment (D): Customer sentiment affects the sales of specific products in each store and is influenced by both advertising and broader economic factors.

Seasonality: Sales experience periodic seasonal trends, with peaks around the holidays and lower numbers during off-seasons.

Trend: There is a general increasing trend in sales as the chain expands.

Noise: Random noise is added to mimic real-world data fluctuations.

Code to Generate Synthetic Time Series Data:

```
import numpy as np
import pandas as pd
import matplotlib.pyplot as plt

# Define time range (e.g., 3 years of daily data)
np.random.seed(42)
days = 365 * 3
time = pd.date_range(start='2020-01-01', periods=days, freq='D')

# Initialize parameters for seasonality, trend, and noise
seasonal_period = 365  # Yearly seasonality
sales_increase_trend = 0.05  # Daily incremental sales growth
noise_level = 0.05  # Noise level

# Define Advertising Spend (A)
advertising_base = 50 + 10 * np.sin(2 * np.pi * np.arange(days)
/ seasonal_period)  # Seasonal ads
advertising_spend = advertising_base +
np.random.normal(scale=5, size=days)  # Add noise to advertising

# Economic factors (C) with a long-term trend and seasonality
economic_factors = 100 + 2 * np.arange(days) / days
+ 10 * np.sin(2 * np.pi * np.arange(days) / (seasonal_period * 2))  # Slow increase and long-
    term seasonality
economic_factors += np.random.normal(scale=3, size=days)  # Add noise to economic factors

# Customer Sentiment (D), influenced by Advertising and Economic factors
customer_sentiment = 70 + 0.3 * advertising_spend + 0.1 * economic_factors
```

```
29  customer_sentiment += np.random.normal(scale=3, size=days)  # Add noise to sentiment
30
31  # Sales (B) is influenced by Advertising,
32  Economic factors, and Customer Sentiment
33  sales_base = 200 + sales_increase_trend * np.arange(days)  # General trend
34  seasonality = 50 * np.sin(2 * np.pi * np.arange(days) / seasonal_period)  # Seasonal variation
35  sales = sales_base + seasonality + 0.4 * advertising_spend
36  + 0.3 * economic_factors + 0.2 * customer_sentiment
37  sales += np.random.normal(scale=noise_level * sales_base, size=days)  # Add noise to sales
38
39  # Create DataFrame to hold the time series data
40  data = pd.DataFrame({
41      'Date': time,
42      'Advertising Spend (A)': advertising_spend,
43      'Economic Factors (C)': economic_factors,
44      'Customer Sentiment (D)': customer_sentiment,
45      'Sales (B)': sales
46  })
47
48  # Set Date as index
49  data.set_index('Date', inplace=True)
50
51  # Display the first few rows of the dataset
52  print(data.head())
53  # Please only save the data here
54  df.to_csv('/Your_data_repo/data.csv', index=False)
```

Explanation for :

A (Advertising Spend) influences B (Sales) directly (1) and D (Customer Sentiment) indirectly (0.5).

B (Sales) directly influences D (Customer Sentiment) (1).

C (Economic Factors) influences both B (Sales) and D (Customer Sentiment) (1).

D (Customer Sentiment) doesn't have a direct influence on other variables, but it affects Sales (B) in real-world scenarios. However, in this matrix, it only affects itself (1).

# D  ADDITIONAL EXPERIMENTS

To evaluate the effectiveness of our proposed TS-Reasoner, we compared it with state-of-the-art reasoning-based models to highlight its advantages in decision-making, compositional optimization, and causal reasoning scenarios. These experiments aim to demonstrate the superior performance of TS-Reasoner across diverse and challenging tasks that require complex reasoning capabilities.

We primarily benchmarked TS-Reasoner against two advanced approaches. The first is o1-preview, an advanced reasoning model developed by OpenAI. o1-preview is specifically designed for tackling tasks requiring multi-step reasoning and decision-making, leveraging large-scale pretraining and structured reasoning pathways to achieve high accuracy. It has demonstrated significant success in tasks requiring complex problem decomposition and logical reasoning, making it an ideal baseline for our evaluation.

The second approach we considered is based on the ReAct framework. This reasoning structure takes inspiration from the dynamic interplay between "reasoning" and "acting," mimicking human behavior when acquiring new skills and solving problems. By integrating reasoning directly into the action process, ReAct is capable of handling tasks requiring adaptive learning and efficient decision-making, which has made it a popular framework for reasoning-based AI systems.

As shown in Table 4,5,10, TS-Reasoner outperforms both baselines in decision-making tasks, compositional QA tasks, as well as causal mining tasks. The experimental result further validated TS-Reasoner as a simple but effective solution to multi-step reasoning in domain specific time series practical application scenarios.

| Task Requirement | TS-Reasoner | | | o1-preview | | | ReAct | | |
|---|---|---|---|---|---|---|---|---|---|
| | SR(%) | AAP | RAP | SR(%) | AAP | RAP | SR(%) | AAP | RAP |
| Profit Percent | **59.2** | **243.31** | **32.34** | 6.1 | 12.53 | -198.43 | 0.0 | - | - |
| Risk Tolerance | **96.0** | 54.54 | -46.04 | 18.0 | **124.72** | **24.14** | 4.0 | -0.04 | -100.63 |
| Budget Allocation | **90.0** | 37.12 | 7.57 | 28.0 | -195.96 | -225.50 | 4.0 | 15.70 | -13.84 |

Table 4: The success rate and performance of TS-Reasoner against additional baselines on desicion making. SR stands for Success Rate; AAP stands for Absolute Average Profit. RAP is the Relative Average Profit compared to vanilla strategy. In Profit Percent and Budget Allocation task, we aim at improving the profit. Thus positive RAP is expected. In Risk Tolerance, the model is required to first ensure the risk and minimize the profit reduction. A negative RAP indicates a more conservative model in terms of risk management compared to vanilla strategy. **Bold** indicates the best results.

| Task | Reasoning Steps | TS-Reasoner | | o1-preview | | ReAct | |
|---|---|---|---|---|---|---|---|
| | | SR(%) | MAPE(std) | SR(%) | MAPE(std) | SR(%) | MAPE(std) |
| Stock Future Price Prediction | 1 | **100.0** | **0.042(0.030)** | 100.0 | 0.053(0.031) | 48.00 | 0.043(0.023) |
| Stock Future Volatility Prediction | 2 | **100.0** | **0.748(0.691)** | 100.0 | 0.750(0.533) | 46.00 | 1.123(0.882) |
| Energy Power w/ Max Load | 3 | **97.87** | 0.101(0.339) | 78.72 | **0.095(0.198)** | 21.28 | 0.136(0.292) |
| Energy Power w/ Min Load | 3 | **97.83** | **0.084(0.104)** | 76.09 | 0.218(0.352) | 36.96 | 0.374(0.796) |
| Load Ramp Rate in Energy Power | 3 | **100.0** | **0.060(0.153)** | 91.67 | 0.076(0.179) | 29.17 | 0.131(0.273) |
| Load Variability Limit in Energy Power | 3 | **93.88** | 0.288 (0.385) | 89.80 | **0.169(0.290)** | 26.53 | 0.268(0.360) |

Table 5: The overall success rate and performance of our model against additional baselines on compositional QA. SR stands for Success Rate; MAPE is the Mean Absolute Percentage Error. **Bold** indicates the best results.

| Task | Reasoning Steps | TS-Reasoner-C | | TS-Reasoner-L | | TS-Reasoner-L + paraphrased data | |
|---|---|---|---|---|---|---|---|
| | | SR(%) | MAPE(std) | SR(%) | MAPE(std) | SR(%) | MAPE(std) |
| Stock Future Price Prediction | 1 | **100.0** | **0.042(0.030)** | **100.0** | **0.042(0.030)** | 20.00 | 0.046(0.030) |
| Stock Future Volatility Prediction | 2 | **100.0** | **0.748(0.691)** | **100.0** | **0.748(0.691)** | 100.0 | 0.748(0.691) |
| Energy Power w/ Max Load | 3 | **97.87** | **0.101(0.339)** | 97.87 | 0.101(0.339) | 97.87 | 0.101(0.339) |
| Energy Power w/ Min Load | 3 | 97.83 | **0.084(0.104)** | 100.00 | 0.086(0.103) | 100.00 | 0.086(0.103) |
| Load Ramp Rate in Energy Power | 3 | **100.0** | 0.060(0.153) | 93.75 | 0.058(0.149) | 97.91 | **0.053(0.144)** |
| Load Variability Limit in Energy Power | 3 | 93.88 | 0.288 (0.385) | **97.96** | **0.203(0.308)** | 89.80 | 0.294(0.375) |

Table 6: The overall success rate and performance of TS-Reasoner variants. TS-Reasoner-C denotes TS-Reasoner with ChatGPT as task decomposer leveraging it's in context learning ability, TS-Reasoner-L denotes TS-Reasoner with finetuned LLAMA as task decomposer, TS-Reasoner-L + paraphrased data denotesTS-Reasoner with LLAMA finetuned on para- phrased data as task decomposer evaluated on paraphrased data. SR stands for Success Rate; MAPE is the Mean Absolute Percentage Error. Bold indicates the best results

# E  VARIANTS OF TS-REASONER

In addition to the original TS-Reasoner (denoted at TS-Reasoner-C), we additionally propose a variant of TS-Reasoner named TS-Reasoner-L with the LLM backbone substituted for Llama 3.1 8b Instruct instead of ChatGPT 3.5 Turbo. Instead of leveraging in context learning ability, we performed finetuning on Llama using the question program pairs in the dataset. As shown in table 6, TS-Reasoner-L performs comparable to TS-Reasoner-C. We further performed token economics analysis as shown in table 9. Although TS-Reasoner-L requires less input tokens compared to TS-Reasoner-C, TS-Reasoner-L incurs the additional computational cost of finetuning. Users can choose the appropriate TS-Reasonerbased on their computational and financial budget.

| Task Requirement | TS-Reasoner | | | o1-preview | | | ReAct | | |
|---|---|---|---|---|---|---|---|---|---|
| | SR(%) | ACC(%) | SSR(%) | SR(%) | ACC(%) | SSR(%) | SR(%) | ACC(%) | SSR(%) |
| Causal Relationship | **100.0** | **79.15** | **8.0** | 82.0 | 74.08 | 4.0 | 50.0 | 76.13 | 0.0 |

Table 7: The success rate and performance of TS-Reasoner against other baselines on causal relationship recognition. SR stands for Success Rate; ACC stands for Accuracy; SSR stands for Strict Success Rate. **Bold** indicates the best results.

| Dataset | Number of CSVs | Avg Total Timestamps | Number of Variables |
|---|---|---|---|
| Daily Yahoo Stock | 6780 | 3785 | 7 |
| Hourly Yahoo Stock | 5540 | 35 | 7 |
| Energy Data | 66 | 872601 | 11 |
| Causal Data | 8 | 529 | 3–6 |

Table 8: Dataset Statistics of the constructed dataset. The exact number of time series are not calculated because it depends on randomly sampled sequence length when generating task instances.

| Task | TS-Reasoner-C | | TS-Reasoner-L | |
|---|---|---|---|---|
| | Avg Input | Avg Output | Avg Input | Avg Output |
| Stock Profit Percent | 2670.0 | 49.0 | 142.0 | 49.0 |
| Stock Risk Tolerance | 2668.0 | 50.6 | 140.0 | 50.6 |
| Stock Budget Allocation | 2676.4 | 66.0 | 148.4 | 66.0 |
| Easy Stock Future Price | 2614.0 | 49.0 | 86.0 | 49.0 |
| Easy Stock Future Volatility | 2609.0 | 41.8 | 81.0 | 41.8 |
| Easy Stock Future Trend | 2613.0 | 45.0 | 85.0 | 45.0 |
| Electricity Prediction Max Load | 2657.6 | 110.6 | 129.6 | 110.6 |
| Electricity Prediction Min Load | 2654.0 | 56.6 | 126.0 | 56.6 |
| Electricity Prediction Load Ramp Rate | 2656.6 | 75.4 | 128.6 | 75.4 |
| Electricity Prediction Load Variability Limit | 2658.6 | 130.0 | 2658.6 | 79.0 |
| Causal Relation | 2648.2 | 74.0 | 120.2 | 74.0 |
| Average | 2647.76 | 63.36 | 119.76 | 63.36 |

Table 9: Token Analysis for each question type. In-Context denotes TS-Reasoner with ChatGPt 3.5 turbo as backbone leveraging its in-context learning ability. Finetuned denotes TS-Reasoner with LLAMA 3.1 8b Instruct finetuned on our dataset as backbone. The total number of input tokens is roughly slightly smaller than number of tokens for system prompt (57) + in-context examples (2424)+ question (119.76) + format instruction (69) = 2669.76. Due to the nature of tokenizers, repetitively occurring phrases may be tokenized as a single token which causes the total number of input tokens to be slightly smaller than the sum of its parts being tokenized individually.

| Task Requirement | w/ Granger | | | w/ Bayesian | | | w/ LiNGAM | | | w/ Causal Forest | | |
|---|---|---|---|---|---|---|---|---|---|---|---|---|
| | SR(%) | ACC(%) | SSR(%) | SR(%) | ACC(%) | SSR(%) | SR(%) | ACC(%) | SSR(%) | SR(%) | ACC(%) | SSR(%) |
| Causal Relationship | **100.0** | **79.15** | 8.0 | **100.0** | 58.61 | 0.0 | **100.0** | 62.10 | 0.0 | 90.0 | 74.81 | **12.0** |

Table 10: The success rate and performance of TS-Reasoner with different causal tools

## F DATASET STATISTICS

Table 8 summarizes the dataset compiled for the multi-step time series reasoning task. The daily yahoo stock prices data involve stock prices for tickers from the earliest available data to September 2024. The hourly yahoo stock price data spans a week in September 2024 with 7 trading hours on each of the 5 business days in a week. The energy data contained electricity load from 6 major electricity grids in the United States (MISO, ERCOT,CAISO,NYISO,PJM, SPP) across 66 zones

for 3 complete years (201 -2020) with a minute level frequency. The electricity load data is paired with corresponding weather variables.

