# OpenReview forum: "Beyond Forecasting: Compositional Time Series Reasoning for End-to-End Task Execution"
_ICLR.cc/2025/Conference — ICLR 2025 Conference Withdrawn Submission_

### Official Review · Reviewer_SXHV · 2024-10-28

**Soundness:** 3
**Presentation:** 2
**Contribution:** 4
**Rating:** 6
**Confidence:** 3

**Summary:**

This work proposes an unified pipeline to solve time-series analysis task. Authors transcribe operations into custom programs and use in-context learning with GPT models to write programs. Authors demonstrate that this approach performs better than some reasoning baselines such as Chain of Thoughts (CoT).

**Strengths:**

1. The problem formulation is both practical and innovative. An unified pipeline for timeseries related task is a significant next step towards advancing timeseries foundation models.
2. Effectively decompose reasoning tasks into steps for streamlined execution and evaluation.

**Weaknesses:**

Method
1. Tree search / Iterative Refinement: Recent work on LLM agents shows that incorporating techniques like tree search or iterative refinement through feedback can enhance an agent's reasoning abilities. It could be useful for the authors to consider implementing these strategies in the future, as they might improve both the compile success rate and overall performance.

Experiment
1. High Standard Deviation in Table. 2: Some rows seem to have very high standard deviation (such as Load Variability Limit in Energy Power). I wonder if this might be due to a small number of evaluation samples or if it’s an aspect of the method itself. It would be helpful to know the number of samples used for each task to better understand these results.

2. Baseline Experiment Details: In the paper, the setup details for baseline experiments are a bit unclear. My concern is whether the baselines have access to the same tool box as TS-Reasoner, such as forecasting models? In addition to access tool boxes, I also wonder if baselines are performed with same number of in-context learning examples used for TS-Reasoner.

3. Token Economics: Since the author used commercial models (GPT) for TS-Reasoner, it would be helpful to provide number of input/output tokens per task evaluated to determine the cost-effectiveness of the pipeline.

**Questions:**

See Weaknesses

---

> ### Author Response · Authors · 2024-11-21
>
> Dear Reviewer: Thank you for your thoughtful and detailed review. We appreciate the time and effort you invested in evaluating our work and providing constructive feedback. Below, we address your concerns and clarify key aspects of our approach.
>
> Q1:  The high standard deviation is actually due to the refinement operation not powerful enough. In most of the cases handling the load variability limit (difference between load power in consecutive timestamps), the refinement op generates code such that it considers the difference of each timestamp’s value with the mean value and cap it at the variability limit in the question. Occasionally, the refinement op generates naive code that considers the difference of each timestamp’s value with zero which led to suboptimal refined results that yielded high MAPE which in turn created large standard deviation. However, we retained the refinement operation as an LLM-generated process to enhance the flexibility of TS-Reasoner, allowing it to automatically parse constraints and generate code, rather than relying on exhaustive, manually written rules and tool specifications.
>
> Q2:  We wanted to clarify that the baselines are not program based approaches. The baselines do not have access to TS-Reasoner’s toolbox nor in context learning samples. We developed the toolbox and in-context learning examples to better serve TS-Reasoner as a program-based decomposer. However, for our most competitive baseline, we enhanced LLM with CoT prompting and additionally allowed it to write code to obtain final results. We posed no restriction in the type of python code it can write. Additionally, although TS-Reasoner only leverages ChatGPT 3.5 turbo, we used ChatGPT 4 turbo for all our baselines to obtain the most competitive result.
>
> Q3: This is a valid point that you raise and we performed token economic analysis for each question type as shown in table 9 attached in the supplementary material pdf. TS-Reasoner does not require any training but requires more input token. On the other hand, we also performed analysis on a variant of TS-Reasoner where LLAMA 3.1 8b Instruct is finetuned on our dataset with question program pairs and served as the backbone of TS-Reasoner instead of ChatGPT 3.5 turbo. The finetuned version of TS-Reasoner is more cost effective when considering inference time cost only but it incurs the additional computational cost of finetuning. And as shown in Table 6, the in-context TS-Reasoner and Finetuned TS-Reasoner achieves similar performance. Users are free to choose one variant over the other with respect to their computational/financial budget.
>
> W1:  We appreciate your suggestion and multi-turn conversation with iterative refinement is indeed one of our future exploration plan!
>
> We sincerely thank you for the insightful comments and constructive feedback, which have helped us improve the clarity and depth of our work. We are happy to engage in further discussions.
>
> |                                     |                     | **TS-Reasoner-C**         |            | **TS-Reasoner-L**       |            | **TS-Reasoner-L+ paraphrased data** |            |
> |-------------------------------------|---------------------|-----------|--------------------|-----------|--------------------|-----------|--------------------|
> | **Task**                            | **Reasoning Steps** | **SR(%)** | **MAPE(std)**      | **SR(%)** | **MAPE(std)**      | **SR(%)** | **MAPE(std)**      |
> | **Stock Future Price Prediction**  | 1                   | **100.0** | **0.042(0.030)**   | **100.0** | **0.042(0.030)**   | 20.00     | 0.046(0.030)       |
> | **Stock Future Volatility Prediction** | 2                | **100.0** | **0.748(0.691)**   | **100.0** | **0.748(0.691)**   | **100.0** | **0.748(0.691)**   |
> | **Energy Power w/ Max Load**       | 3                   | **97.87** | **0.101(0.339)**   | **97.87** | **0.101(0.339)**   | **97.87** | **0.101(0.339)**   |
> | **Energy Power w/ Min Load**       | 3                   | 97.83     | **0.084(0.104)**   | **100.00** | 0.086(0.103)      | **100.00** | 0.086(0.103)       |
> | **Load Ramp Rate in Energy Power** | 3                   | **100.0** | 0.060(0.153)       | 93.75     | 0.058(0.149)       | 97.91     | **0.053(0.144)**   |
> | **Load Variability Limit in Energy Power** | 3            | 93.88     | 0.288(0.385)       | **97.96** | **0.203(0.308)**   | 89.80     | 0.294(0.375)       |
>
> Table 6: The overall success rate and performance of TS-Reasoner variants. TS-Reasoner-C denotes TS-Reasoner with ChatGPT as task decomposer leveraging it's in context learning ability, TS-Reasoner-L denotes TS-Reasoner with finetuned LLAMA as task
> decomposer, TS-Reasoner-L  + paraphrased data denotesTS-Reasoner with LLAMA finetuned on para-
> phrased data as task decomposer evaluated on paraphrased data. SR stands for Success Rate; MAPE
> is the Mean Absolute Percentage Error. Bold indicates the best results

---

> ### Author Response · Authors · 2024-11-24
>
> We are attaching the analysis in markdown format for your convenience
>
> | |TS-Reasoner-C| |TS-Reasoner-L ||
> |-----------------------------------------|----------------------------|-----------------------------|---------------------------|----------------------------|
> | **Task**                                | **Avg Input** | **Avg Output** | **Avg Input** | **Avg Output** |
> | Stock Profit Percent                    | 2670.0                    | 49.0                        | 142.0                     | 49.0                       |
> | Stock Risk Tolerance                    | 2668.0                    | 50.6                        | 140.0                     | 50.6                       |
> | Stock Budget Allocation                 | 2676.4                    | 66.0                        | 148.4                     | 66.0                       |
> | Easy Stock Future Price                 | 2614.0                    | 49.0                        | 86.0                      | 49.0                       |
> | Easy Stock Future Volatility            | 2609.0                    | 41.8                        | 81.0                      | 41.8                       |
> | Easy Stock Future Trend                 | 2613.0                    | 45.0                        | 85.0                      | 45.0                       |
> | Electricity Prediction Max Load         | 2657.6                    | 110.6                       | 129.6                     | 110.6                      |
> | Electricity Prediction Min Load         | 2654.0                    | 56.6                        | 126.0                     | 56.6                       |
> | Electricity Prediction Load Ramp Rate   | 2656.6                    | 75.4                        | 128.6                     | 75.4                       |
> | Electricity Prediction Load Variability Limit | 2658.6              | 130.0                       | 2658.6                    | 79.0                       |
> | Causal Relation                         | 2648.2                    | 74.0                        | 120.2                     | 74.0                       |
> | **Average**                             | **2647.76**               | **63.36**                   | **119.76**                | **63.36**                  |
>
> Table 9: Token Analysis for each question type. TS-Reasoner-C denotes TS-Reasoner with ChatGPt 3.5
> turbo as backbone leveraging its in-context learning ability. TS-Reasoner-L denotes TS-Reasoner with
> LLAMA 3.1 8b Instruct finetuned on our dataset as backbone. The total number of input tokens
> is roughly slightly smaller than number of tokens for system prompt (57) + in-context examples
> (2424)+ question (119.76) + format instruction (69) = 2669.76. Due to the nature of tokenizers,
> repetitively occurring phrases may be tokenized as a single token which causes the total number of
> input tokens to be slightly smaller than the sum of its parts being tokenized individually.

---

> > ### Comment · Reviewer_SXHV · 2024-11-26
> > **Response to discussion**
> >
> > Thank you for your time addressing my concerns. I decided to keep the current score and the opinion that this work has positive contribution to the community.

---

### Official Review · Reviewer_3VA4 · 2024-10-29

**Soundness:** 1
**Presentation:** 1
**Contribution:** 1
**Rating:** 1
**Confidence:** 5

**Summary:**

The paper introduces the "Compositional Time Series Reasoning Task". It proposes a TS-Reasoner model that uses an ChatGPT-4-turbo and time series models to create time series forecasting pipelines based on 1) question, 2) program, 3) output, 4) evaluation. While this is a creative concept, the overall contribution seems to lack depth and rigor in advancing either the NLP or time series forecasting fields.

The paper would benefit from a clearer focus on advancing state-of-the-art methodologies, rather than merely leveraging LLMs for the sake of "novelty".

**Strengths:**

Somehow interesting ChatGPT-4-turbo prompt engineering for time series analysis.

**Weaknesses:**

1. The paper presents a caricature of a time series analysis task. Their examples on Figure 1 are trivial tasks.

2. The process of arbitrarily requesting the ChatGPT-4-turbo to predict stock prices and receiving an ARIMA output is highly caricaturesque. This illustrates a superficial application of time series models, where the task of forecasting is treated trivially. Similarly, the request for a causal analysis that results in a standard Granger causality output reinforces this pattern of overly simplistic and mechanical responses. These examples undermine the paper's claim to tackle complex reasoning tasks, as the system is merely pairing standard methods with generic prompts, without demonstrating true depth or sophistication in either reasoning or time series analysis.

**Questions:**

1. The evaluation of the proompt engineering tool is poorly defined. How can an LLM tuned for complex time series reasoning be evaluated with Profit Percent, Risk Tolerance, and budget allocation. What are those metrics measuring, is the model being evaluated on a simulator, or is the model taking decisions on real scenarios? Everything is so high level and unclear.

2. Experiments are mostly limited to making calls to ChatGPT-4-turbo. Is this a real submission to ICLR?

---

> ### Author Response · Authors · 2024-11-21
>
> Dear Reviewer: Thank you for your review. We would like to point out politely that there are major confusions regarding the contribution of our work. Below we hope to clarify key aspects of our approach:
>
> W1: In Figure 1, we provide a comprehensive depiction of time series analysis tasks, including numerical prediction, trend evaluation, investment decision-making, combinatorial optimization, and causal inference. These tasks encompass the most mainstream time series scenarios with real world applications, we do not believe that these overly simplistic time series analysis tasks.
>
> W2: We would like to clarify this misunderstanding. Our work does not involve arbitrarily requesting ChatGPT-4-turbo to use ARIMA for time series forecasting. What you referred to is our baseline. Specifically, this baseline employs prompts to reasonably instruct ChatGPT-4-turbo to independently analyze the characteristics of time series and utilize appropriate models for forecasting. The experimental results demonstrate that TS-Reasoner method achieves significantly better performance, which fully validates the effectiveness of our approach. In fact, our method leverages in-context learning to guide the model in automatically identifying the characteristics of different time series tasks, effectively decomposing and executing complex tasks. This enables TS-Reasoner to achieve accurate reasoning with superior performance. In contrast, neither standalone ChatGPT-4-turbo nor ChatGPT-4-turbo enhanced with CoT can achieve comparable effectiveness. We have further illustrated in table 10 as attached in the supplementary material pdf that Granger Causality is not overly simplistic and actually demonstrates superior performance among causal models that support direct inference.
>
> Q1:  We believe this is a misunderstanding. Profit Percent, Risk Tolerance, and Budget Allocation are constraints from questions that users can specify rather than metrics. With these types of constraints we form question instances. The metrics we used are success rate and average profit. We do not evaluate LLM itself on profit percent, risk tolerance, or budget allocation and hence no simulator is involved. For standalone LLM models (e.g., the powerful ChatGPT-4-turbo), when stock information and user investment requirements are provided as inputs, it is very difficult for the model to generate a reasonable stock portfolio that satisfies the user's investment constraints. We evaluate the final result of TS-Reasoner  (obtained from sequential execution of programmatic tools outlined by the task decomposer) on success rate and average profit. The final output is not text but rather structured numerical output from modules in the toolbox. For example, given an investment strategy containing the allocation of money among stocks of interest and the optimized buy and sell date, we can apply this strategy to the ground truth future data to calculate the profit obtained by this strategy and checking whether constraints like budget are violated as we have the ground truth stock price. We are only asking questions that the data can answer.
>
> Q2: Please refer to the response to Question 1. We sincerely appreciate your review and feedback. However, it appears that there may be some misunderstanding of the core ideas of our work, and we genuinely hope that our response will be beneficial to clearing things up. TS-Reasoner automates the practical time series analysis processes in real world applications rather than inventing new processes  and it serves as a practical tools for domain-specific tasks.
>
> We are attaching the additional experiment result tables in markdown format for your convenience.
>
> |                         | **w/ Granger**         |            |            | **w/ Bayesian**      |            |            | **w/ LiNGAM**         |            |            | **w/ Causal Forest**  |            |            |
> |-------------------------|-----------|------------|------------|-----------|------------|------------|-----------|------------|------------|-----------|------------|------------|
> | **Task Requirement**   | **SR(%)** | **ACC(%)** | **SSR(%)** | **SR(%)** | **ACC(%)** | **SSR(%)** | **SR(%)** | **ACC(%)** | **SSR(%)** | **SR(%)** | **ACC(%)** | **SSR(%)** |
> | **Causal Relationship** | **100.0** | **79.15**   | 8.0        | **100.0** | 58.61      | 0.0        | **100.0** | 62.10      | 0.0        | 90.0      | 74.81      | **12.0**    |
>
> Table 10: The success rate and performance of TS-Reasoner with different causal tools

---

> > ### Comment · Reviewer_3VA4 · 2024-11-25
> > **W1: On the over simplification of the time series analysis tasks**
> >
> > The application of large language models (LLMs) to time series analysis, like numerical prediction, trend evaluation, investment decision-making, combinatorial optimization, and causal inference, is fundamentally limited by their reliance on API calls to basic algorithms. These tasks require domain-specific expertise and robust, specialized methodologies, which typically necessitate dedicated teams of experts within organizations. Simplifying such processes to an LLM acting as an intermediary is both reductive and misrepresents the capacity of these tools.
> >
> > Moreover, suggesting that LLMs can effectively handle critical tasks such as "investment decision-making" raises serious concerns. A key question for the authors is this: Would you trust your money to a system built on the described approach, allowing it to make investment decisions autonomously? If the answer is no, advocating for such applications seems premature and potentially misleading.

---

> > > ### Comment · Reviewer_3VA4 · 2024-11-25
> > > **W2: On the the lack of a proper validation of the effectiveness of the TS-Reasoner approach.**
> > >
> > > The TS-Reasoner approach is far from fully validated. Specifically, the results in Table 7 for Stock Future Price/Volatility Prediction, which narrowly focuses on forecasting Yahoo stock daily/hourly price series, raise concerns. The forecast accuracy is within error bands of the ChatGPT baseline, and the evaluation is done a posteriori on a single series. Table 7's limited scope is more likely suited for a demonstration than a rigorous scientific evaluation.
> > >
> > > In financial forecasting, meaningful validation requires extensive testing across diverse datasets, markets, and conditions, accounting for inherent noise and variability. Additionally, reporting marginal improvements without clear statistical significance or robust benchmarking against state-of-the-art models raises doubts about the practical impact of your approach.

---

> > > > ### Comment · Reviewer_3VA4 · 2024-11-25
> > > > **W2: On the the lack of a proper validation of the effectiveness of the TS-Reasoner approach.**
> > > >
> > > > Predicting electricity demand is a critical and highly complex task for energy management agencies, with significant operational, economic, and societal implications. Overestimating electricity demand can lead to unnecessary generation, which not only wastes resources but also increases costs, reduces system efficiency, and, in extreme cases, causes equipment damage. Conversely, underestimating demand can result in insufficient supply, leading to blackouts or the need for emergency interventions—both of which are costly and disruptive to consumers and businesses alike.
> > > >
> > > > Given the stakes, electricity forecasting is a mature and specialized field supported by dedicated journals and extensive research into advanced methodologies. In this context, an anecdotal evaluation claiming that TS-Reasoner improves on ChatGPT forecasts for a narrow set of tasks fails to meet the rigorous standards expected in the field.

---

> > > > > ### Author Response · Authors · 2024-12-03
> > > > >
> > > > > We are not replacing the research on electricity forecasting, we are automating the process and incorporating system constraints. Additionally, MAPE is a very common and valid metric used in load forecasting.

---

> > > > ### Author Response · Authors · 2024-12-03
> > > >
> > > > For our forecasting task, we are indeed using state-of-the-art time series model TEMPO selected for its strong zero shot forecasting ability. TS-Reasoner is not a forecasting model but a framework for automating complex task workflow.

---

> > > ### Author Response · Authors · 2024-12-03
> > >
> > > This is indeed our goal, and we have shown experimental result in Table 1 of the paper that TS-Reasoner demonstrates success towards this goal.

---

> > ### Comment · Reviewer_3VA4 · 2024-11-25
> > **W2: On the the lack of a proper validation of the effectiveness of the TS-Reasoner approach.**
> >
> > Several concerns arise regarding the evaluation of Granger causality using TS-Reasoner:
> >
> > 1. Was the data tested for stationarity, or is TS-Reasoner potentially misinterpreting cointegrated series?
> > 2. Are the relationships between variables linear, as Granger causality is designed for linear dependencies in VAR models?
> > 3. Are the errors of the implied VAR model Normally distributed? Without this, the test risks misspecification issues.
> > 4. What is the coverage of multivariate series that can be correctly modeled with VARs and Normality error assumptions?
> > 5. Are the relationships between the variables constant over time?
> >
> > How does TS-Reasoner address these challenges? What happens if a TS-Reasoner user faces any of the errors above?

---

> > > ### Author Response · Authors · 2024-12-03
> > >
> > > Based on the review, we have already updated experimental results with different causal models. These questions are not in the realm of this paper to address.

---

> > ### Comment · Reviewer_3VA4 · 2024-11-25
> > **Q1: On the lack of clarity of TS-Reasoner stock portfolio creation capabilities**
> >
> > The paper lacks a proper literature review on stock portfolio creation.
> >
> > What algorithm does TS-Reasoner use to create stock portfolios? I encourage the authors to explicitly confirm that TS-Reasoner is the model responsible for portfolio creation, as it currently seems to misattribute this task to TS-Reasoner.
> >
> > Why is the paper "Income hedging and portfolio decisions" (Bonaparte et al., 2014, https://econpapers.repec.org/article/eeejfinec/v_3a113_3ay_3a2014_3ai_3a2_3ap_3a300-324.htm) cited as the source for stock portfolio creation? This paper estimates Dutch households' risk propensity.

---

> > > ### Author Response · Authors · 2024-12-03
> > >
> > > We are answering basic questions regarding investment, this is not a portfolio management paper.

---

> ### Comment · Reviewer_3VA4 · 2024-11-25
> **Q2: Clarification of ChatGPT-4 turbo used as baseline for time series analysis and forecasting tasks**
>
> It perplexing that we are debating the validity of using ChatGPT-4 turbo as a forecasting baseline.

---

> > ### Author Response · Authors · 2024-12-03
> >
> > It's not exactly clear what your question is but we did include chatgpt-4-turbo results in table 2 of the paper.

---

### Official Review · Reviewer_b7iA · 2024-10-31

**Soundness:** 1
**Presentation:** 3
**Contribution:** 2
**Rating:** 5
**Confidence:** 4

**Summary:**

This paper introduces a novel task in time series analysis for the financial and energy domains, termed "Compositional Time Series Reasoning," designed to address complex, multistep reasoning tasks that integrate time series data with domain knowledge for applications like decision-making and compositional question-answering. Traditional time series tasks (e.g., forecasting, classification, and anomaly detection) emphasize single-task predictive accuracy, whereas the proposed task requires synthesis across multiple reasoning steps. To tackle this challenge, the authors present TS-Reasoner, a program-aided approach that leverages large language models to decompose complex tasks into sequential steps of programs. TS-Reasoner supports the creation of custom modules and can integrate domain knowledge and user-specified constraints, offering flexibility in reasoning tasks. The paper validates TS-Reasoner’s effectiveness on new datasets within finance and energy domains, showcasing improved performance across a series of question types and reasoning requirements.

**Strengths:**

**S1:** The paper introduces a novel task in time series analysis, where step-by-step reasoning with large language models (LLMs) for compositional reasoning is an innovative approach.

**S2:** The accompanying video aids readers in understanding the proposed methodology.

**S3:** The introduction clearly presents relevant concepts, situates the paper’s contributions, and effectively outlines the paper's positioning within the field.

**Weaknesses:**

**W1:** The technical contribution of the paper appears limited, as the core technique mainly involves adapting Chain-of-Thought (CoT) reasoning to time series tasks without offering any significant advancements.

**W2:** The research is confined to financial and energy domains, and it’s unclear how the proposed method could be adapted to other time series domains, such as weather or traffic. While it may not be necessary for the authors to demonstrate explicit adaptations, the approach lacks generality and seems overly dependent on prompt design.

**W3:** Using LLMs for compositional reasoning in time series tasks presents various challenges, including understanding time series data in text form, selecting appropriate models based on that understanding, executing the models, and verifying the results. These challenges are not discussed or resolved in the paper; the authors instead rely solely on CoT and simple prompts to guide the LLM, with phrases like “Understand and parse the input data from text” (Line 970) and “Choose an appropriate model… Make sure the model is suitable” (Line 971).

**W4:** The functionality of the Task Decomposer is critical, as it involves guiding the LLM in understanding time series and making decisions based on that understanding. However, the authors do not clearly present how this module operates, especially in terms of how the three modules are coordinated.

**W5:** The paper lacks a sufficient number of baselines, including only two, which is too few. Other relevant baselines, such as various time series forecasting models (e.g., statistical models, transformer-based models) and other in-context learning models (e.g., REACT), should be included for a more comprehensive comparison.

**Questions:**

**Regarding the Methodology:**
- In Line 363, the paper claims that "Time Series Model Modules are grounded in foundation time series models." However, in Appendix C.3, the prompts reveal that the final predictions utilize models like ARIMA and LSTM.

- For model selection, the prompt used by the authors is “Choose an appropriate model: You will select an appropriate time series forecasting model (e.g., ARIMA, LSTM, etc.) based on the input data.” This approach raises two concerns. Firstly, it contradicts the statement about grounding in foundation models (line 363). Secondly, I question the effectiveness of such prompts. As a simple example, could the authors demonstrate which models were ultimately selected, and in what proportions? Based on my experience, the model likely defaults to ARIMA in most cases due to inherent biases in LLMs rather than a true understanding of the data.

**Regarding the Experiments:**

- Although the authors design different tasks and metrics specifically for financial and energy domains, like absolute average profit (AAP) and relative average profit (RAP), I question the effectiveness of the proposed model in handling these tasks. For instance, in line 275, the authors prompt the LLM with specific requirements like: “[1. I want to make at least {profit percent}% profit. 2. I have a risk tolerance of {risk percent}%.” It’s unclear whether the LLM can genuinely understand and resolve these requirements. The omission of a substantial portion of baselines makes it difficult to find evidence supporting the effectiveness of the proposed method.

- For decision-making tasks, the authors omit other key evaluation metrics, such as MSE and RMSE, which are essential for gauging model performance.

---

> ### Author Response · Authors · 2024-11-21
>
> Dear Reviewer: Thank you for your thoughtful and detailed review. We appreciate the time and effort you invested in evaluating our work and providing constructive feedback. Below, we address your concerns and clarify key aspects of our approach.
>
> W1: We would like to clarify that our **primary contribution lies in empowering LLM to handle complex, multi-step time series reasoning through program-aided decomposition.** Out approach does not use CoT or extend CoT, but we considered CoT as a baseline method as it is one of the mainstream techniques for general LLM Reasoning. Unlike CoT, which focuses on sequential natural language reasoning, our framework integrates **structured programmatic reasoning**, bridging the gap between natural language understanding and numerical computation in time series tasks. This hybrid approach enables the model to handle diverse tasks that involve both domain knowledge and data-specific constraints, as evidenced by our results in financial decision-making and energy load forecasting tasks.
>
> W2: We note that the underlying program-based reasoning approach and modular design are inherently flexible and can be adapted to new domains with minimal changes, primarily by defining relevant domain-specific modules and crafting question instances as well as its evaluation protocol. We want to mention that developing question specific evaluation protocol is unavoidable regardless of the type of system you design as complex questions are diverse in nature and the evaluation protocol should be specific to question types. We would also like to mention that when extending to other domains where the question type remains unchanged such as predicting traffic flow from time 1 to T and knowing that the maximum capacity of the road is x at any given time, then no adaptation is required on TS-Reasoners end except for adding to in-context examples. The new question essentially still requires a forecasting procedure followed by a refinement procedure. In summary, minimal effort is required for extending to other domain’s data but some more effort needs to be put into creating tasks instances and evaluation protocols if we are adapting to other domain specific question types. This design makes TS-Reasoner highly flexible and adaptable.
> In addition, we want to emphasize our design philosophy of creating a practical tool for real world scenarios time series analysis for domain experts rather than general purpose systems that may lack precision of task specific understanding as evidenced by our original experimental results and additional comparison with o1 and ReAct baseline shown in table 4,5,7 in the supplementary material pdf (standalone general purpose LLMs can perform these tasks but demonstrate poor performance) . Moreover, we believe we are the first to propose the task definition of compositional time series reasoning and would like to draw attention for researchers in the field to work on such practical scenarios that we collected from surveying domain literatures[1][2][3]. Lastly, in reality, scientists and engineers only have limited questions they are interested in asking in the domain which TS-Reasoner is perfectly suitable for.
>
> [1] D’Amico, Guglielmo, Filippo Petroni, and Salvatore Vergine. "Ramp rate limitation of wind power: An overview." Energies 15, no. 16 (2022): 5850.
>
> [2] Zidan, Aboelsood, and Ehab F. El-Saadany. "Distribution system reconfiguration for energy loss reduction considering the variability of load and local renewable generation." Energy 59 (2013): 698-707.
>
> [3] Lee, Tae Kyun, Joon Hyung Cho, Deuk Sin Kwon, and So Young Sohn. "Global stock market investment strategies based on financial network indicators using machine learning techniques." Expert Systems with Applications 117 (2019): 228-242.
>
> W3: We would like to clarify this misunderstanding. CoT is not part of our methodology and is only used as baselines. The prompts you mentioned in line 970 is in section C.3 which is CoT prompts, prompts used for TS-Reasoner is in section C.1 titled TS-Reasoner prompts. We include in context examples, question, instructions for learning from in context examples. TS-Reasoner does not process time series in text format as we believe the development of time series foundation models trained a large time series dataset is more suitable to handle time series data than LLM whether it’s pure pretrained LLM of finetuned LLM. In fact, we do not contradict the statement about grounding in foundation models (line 363). We chose the programmatic approach so that this hybrid design allows LLM to understand the task scenarios while allowing modules in the toolbox to handle sophisticated subtasks such as making forecast and making numerical refinement. We believe we are the first to propose this approach in tacking compositional reasoning in time series.

---

> ### Author Response · Authors · 2024-11-21
>
> W4: The task decomposer’s responsibility is to understand the task instance and decompose the task into smaller subtasks that can be tackled by modules in the toolbox. There are 11 modules in our toolbox but are generally grouped into three categories as shown in Figure 2. The task decomposer can use any modules in the toolbox and order them in any way according to its understanding of the task and previous examples. Our program executor takes the output of task decomposer which is a string that consists of the execution plan and calls the corresponding module in the toolbox with the correct arguments. Final result is obtained after execution completes.
>
> W5: We conducted comparative experiments between  TS-Reasoner and the ReAct reasoning framework across three key tasks: decision-making, compositional QA, and causal relationship recognition. The results are attached in the supplementary material pdf.
> As shown in Table 4, our method significantly outperforms ReAct in all three decision-making tasks Profit Percent, Risk Tolerance, and Budget Allocation. Through case analysis, we found that while ReAct demonstrates strong capabilities in integrating reasoning, acting, and observation, it struggles in complex decision-making tasks. Specifically, ReAct encounters difficulties in accurately decomposing and executing detailed decision-making steps, often leading to suboptimal or incorrect actions with poor success rates. In contrast, TS-Reasoner excels in these scenarios by effectively breaking down complex tasks and ensuring precise execution, highlighting its superior ability to handle intricate decision-making problems.
> In compositional QA tasks, the results presented in Table 5 show that TS-Reasoner consistently outperforms ReAct across various tasks. For simple reasoning tasks such as Stock Future Price Prediction and Volatility Prediction, as well as constraint-based complex reasoning tasks, our method delivers significantly better performance.
> In the causal relationship recognition task, as illustrated in Table 7, TS-Reasoner outperforms ReAct in both success rate and accuracy. This underscores TS-Reasoner's strength in accurately identifying causal dependencies and reasoning about them, a critical requirement for many real-world applications.
> Overall, even when compared with ReAct, which integrates reasoning, acting, and observation into its reasoning pathways, TS-Reasoner delivers superior performance in nearly all tasks. While ReAct is better suited for tasks such as information retrieval and external knowledge augmentation, it is less effective in scenarios requiring accurate compositional reasoning and precise time series understanding and predictions. These experiments validate the superiority of TS-Reasoner as a domain specific reasoning framework for time series tasks, establishing its effectiveness and versatility across diverse and challenging tasks.
>
> Q1,Q2: Please see W3.
>
> Q3:  We only ask the task decomposer to properly understand and decompose the task. For the stock investment scenario, the task decomposer should learn that the task needs to be decomposed to predicting future stock price, creating the objective, creating the constraint, and pass them all to the optimization tool and obtain an investment strategy. The optimization of maximizing profit while considering the constraint is handled by the optimization module which gets passed the correct arguments such as the profit percent by the task decomposer (LLM backbone).
>
> Q4:  For decision-making in financial domains, people care about portfolio gain so we used profit gain to measure performance. The output of portfolio optimization task is an investment strategy containing the allocation of money among stocks of interest and the optimized buy and sell date. MSE and MAE are inapplicable in this case. If you are referring to forecasting performance of our forecasting tool, we show MAPE in the stock price forecasting task and MAPE is chosen because it is data-scale independent.
>
> We sincerely thank you for the insightful comments and constructive feedback, which have helped us improve the clarity and depth of our work. We are happy to engage in further discussions.

---

> ### Author Response · Authors · 2024-11-24
>
> We are attaching the additional experimental result tables in markdown format for your convenience.
>
> |                       | **TS-Reasoner**     |           |           | **o1-preview**    |           |            | **ReAct**   |           |             |
> |-----------------------|-----------|-----------|-----------|-----------|-----------|------------|-----------|-----------|-------------|
> | **Task Requirement** | **SR(%)** | **AAP**   | **RAP**   | **SR(%)** | **AAP**   | **RAP**    | **SR(%)** | **AAP**   | **RAP**     |
> | **Profit Percent**    | **59.2**  | **243.31**| **32.34** | 6.1       | 12.53     | -198.43    | 0.0       | -         | -           |
> | **Risk Tolerance**    | **96.0**  | 54.54     | -46.04    | 18.0      | **124.72**| **24.14**  | 4.0       | -0.04     | -100.63     |
> | **Budget Allocation** | **90.0**  | **37.12** | **7.57**  | 28.0      | -195.96   | -225.50    | 4.0       | 15.70     | -13.84      |
>
> Table 4: The success rate and performance of TS-Reasoner against additional baselines on desicion
> making. SR stands for Success Rate; AAP stands for Absolute Average Profit. RAP is the Relative
> Average Profit compared to vanilla strategy. In Profit Percent and Budget Allocation task, we aim at
> improving the profit. Thus positive RAP is expected. In Risk Tolerance, the model is required to first
> ensure the risk and minimize the profit reduction. A negative RAP indicates a more conservative
> model in terms of risk management compared to vanilla strategy. Bold indicates the best results.
> We are attaching the markdown version of the new tables for your convenience.
>
> |                                     |                     | **TS-Reasoner**      |            | **o1-preview**      |            | **ReAct**          |                    |
> |-------------------------------------|---------------------|-----------|--------------------|-----------|--------------------|-----------|--------------------|
> | **Task**                            | **Reasoning Steps** | **SR(%)** | **MAPE(std)**      | **SR(%)** | **MAPE(std)**      | **SR(%)** | **MAPE(std)**    |
> | **Stock Future Price Prediction**  | 1                   | **100.0** | **0.042(0.030)**   | 100.0     | 0.053(0.031)       | 48.00     | 0.043(0.023)       |
> | **Stock Future Volatility Prediction** | 2                | **100.0** | **0.748(0.691)**   | 100.0     | 0.750(0.533)       | 46.00     | 1.123(0.882)       |
> | **Energy Power w/ Max Load**       | 3                   | **97.87** | 0.101(0.339)       | 78.72     | **0.095(0.198)**   | 21.28     | 0.136(0.292)       |
> | **Energy Power w/ Min Load**       | 3                   | **97.83** | **0.084(0.104)**   | 76.09     | 0.218(0.352)       | 36.96     | 0.374(0.796)       |
> | **Load Ramp Rate in Energy Power** | 3                   | **100.0** | **0.060(0.153)**   | 91.67     | 0.076(0.179)       | 29.17     | 0.131(0.273)       |
> | **Load Variability Limit in Energy Power** | 3            | **93.88** | 0.288(0.385)       | 89.80     | **0.169(0.290)**   | 26.53     | 0.268(0.360)       |
>
> Table 5: The overall success rate and performance of our model against additional baselines on
> compositional QA. SR stands for Success Rate; MAPE is the Mean Absolute Percentage Error.
> Bold indicates the best results.
> |                         | **TS-Reasoner**     |            |            | **o1-preview**     |            |            | **ReAct**   |            |            |
> |-------------------------|-----------|------------|------------|-----------|------------|------------|-----------|------------|------------|
> | **Task Requirement**   | **SR(%)** | **ACC(%)** | **SSR(%)** | **SR(%)** | **ACC(%)** | **SSR(%)** | **SR(%)** | **ACC(%)** | **SSR(%)** |
> | **Causal Relationship** | **100.0** | **79.15**   | **8.0**    | 82.0      | 74.08      | 4.0        | 50.0      | 76.13      | 0.0        |
>
> Table 7: The success rate and performance of TS-Reasoner against other baselines on causal relationship recognition. SR stands for Success Rate; ACC stands for Accuracy; SSR stands for Strict
> Success Rate. Bold indicates the best results.

---

> ### Comment · Reviewer_b7iA · 2024-11-25
>
> Thank you for your detailed response to my review. While I appreciate your efforts to address the concerns raised, there are still several critical points that require further clarification:
>
> 1. **Regarding Technical Innovation**
>     - Your response claims that your approach is fundamentally different from CoT, but the distinction remains unclear. While you integrate programmatic components, the core reasoning process still relies heavily on LLM's ability to decompose tasks through prompting. Please clarify:
>         - What specific technical innovations distinguish your approach from existing program-aided reasoning frameworks?
>         - How does your method handle the inherent limitations of LLMs in numerical reasoning?
>
> 2. **Regarding Generalizability**
>     - While you argue that the framework is adaptable, the response doesn't fully address the concern about domain dependence. Specifically:
>         - The claim that "minimal effort is required for extending to other domain's data" needs empirical validation.
>         - The cited examples (traffic flow) are structurally similar to your test cases.
>         - How would the system handle fundamentally different types of time series tasks?
>
> 3. **Regarding Model Selection**
>     - Your response about not contradicting the foundation model statement is not convincing:
>         - The paper claims to use foundation time series models (Line 363), but the implementation relies on traditional models like ARIMA and LSTM. This question hasn't been addressed yet.
>         - There's no clear explanation of how the system selects appropriate models for different scenarios.
>         - Please provide empirical evidence of model selection effectiveness.
>
> 4. **Regarding Baseline Comparisons**
>     - While you've added ReAct as a baseline, several concerns remain:
>         - The comparison still lacks standard time series models.
>         - The evaluation focuses only on end-task metrics without analyzing intermediate reasoning steps.
>         - The success criteria for different tasks need a more rigorous definition.
>
> 5. **Regarding Evaluation Metrics**
>     - Your justification for not using MSE/RMSE in forecasting components needs reconsideration:
>         - While profit-based metrics are relevant for final decisions, intermediate forecasting accuracy should still be evaluated using standard metrics.
>         - This would help isolate the performance of different components in your pipeline.

---

> ### Author Response · Authors · 2024-11-29
>
> Dear Reviewer, thank you for response
>
> 1. LLM is performing task decomposition reasoning in TS-Reasoner instead numerical reasoning. Numerical time series data is handled by time series models and other numerical routines and this is exactly the reason why we opted for this hybrid approach of integrating LLM with numerical tools and time series models. We are a program-aided approach but we additionally support custom LLM generation modules to handle user constraints which is more flexible than exhausting all possible constraints and writing a module to handle each constraint.
>
> 2. The traffic example was given for adaptation to other domain data only because many important scientific questions present the same form. For adapting other types of questions, we need to equip TS-Reasoner with relevant tools and provide example usage of such tools in the in-context examples. Scientists and engineers usually have a limited number of questions they are interested in asking. We are happy to hear questions that you have in mind and we are open to running more experiments on questions you propose.
>
> 3. We used the time series foundation model TEMPO chosen for its strong zero shot forecasting ability. We allowed our baseline models to select models to their discretion to get the most competitive resultOur baseline reasoning approach ChatGPT chooses to use ARIMA. We did not claim TS-Reasoner performs model selection at this stage.
>
> 4. ReAct does not have access to time series models and always reverts back to the default tool. ReAct also does not have any capability of writing code such as creating a pandas dataframe. Additionally ReAct demonstrates a strong pattern of repeating. ReAct has the ability to retrieve relevant information from external database and was tested for QA and fact verification in the original paper which are the tasks that it was more suitable for.  In summary, ReAct does not possess the ability to perform time series analysis tasks.  See examples below
> Success Criteria: the model outputs the final result in the correct format (a list or a numpy array or a dictionary as specified in the question), and all constraints are satisfied. The constraints are domain specific, it could be maximum allowable system load in an electric grid or risk tolerance of investment.
>
> 5. Yes, we have additionally included the intermediate evaluation metrics.  Please note that the RMSE values are dependent on the data scale and values of stock prices vary by question which is the reason why we chose MAPE as our metric originally.
>
>
> Budget allocation
>
> RMSE: 1.7700392574630928 (2.639736897452789)
>
> Risk Tolerance
>
> RMSE: 1.4887427890030331 (1.515834108214604)
>
> Profit Percent
>
> RMSE: 3.826622980068753(8.813945680556516)
>
> We are very grateful for your feedback.

---

> ### Author Response · Authors · 2024-11-29
>
> ReAct example:
> To create an investment strategy, I need to analyze the historical stock data provided for each stock, predict their future trends, and calculate the optimal number of units to buy, the best time to buy them, and how long to hold them to maximize the expected profit within the given budget and risk tolerance.
>
> Action: I will first calculate the average price and the standard deviation (volatility) for each stock based on the historical data provided. This will help in understanding the general price movement and the risk associated with each stock.
>
> Action Input: Calculate the mean and standard deviation for each stock using the historical data provided.
>
> Observation: I will first calculate the average price and the standard deviation (volatility) for each stock based on the historical data provided. This will help in understanding the general price movement and the risk associated with each stock. is not a valid tool, try one of [Calculator].
>
> Thought:To create an investment strategy, I need to analyze the historical stock data provided for each stock, predict their future trends, and calculate the optimal number of units to buy, the best time to buy them, and how long to hold them to maximize the expected profit within the given budget and risk tolerance.
>
> Action: I will first calculate the average price and the standard deviation (volatility) for each stock based on the historical data provided. This will help in understanding the general price movement and the risk associated with each stock.
>
> Action Input: Calculate the mean and standard deviation for each stock using the historical data provided.
>
> Observation: I will first calculate the average price and the standard deviation (volatility) for each stock based on the historical data provided. This will help in understanding the general price movement and the risk associated with each stock. is not a valid tool, try one of [Calculator].
>
> Thought:To create an investment strategy, I need to analyze the historical stock data provided for each stock, predict their future trends, and calculate the optimal number of units to buy, the best time to buy them, and how long to hold them to maximize the expected profit within the given budget and risk tolerance.
>
> Action: I will first calculate the average price and the standard deviation (volatility) for each stock based on the historical data provided. This will help in understanding the general price movement and the risk associated with each stock.
>
> Action Input: Calculate the mean and standard deviation for each stock using the historical data provided.
>
> Observation: I will first calculate the average price and the standard deviation (volatility) for each stock based on the historical data provided. This will help in understanding the general price movement and the risk associated with each stock. is not a valid tool, try one of [Calculator].
>
> Thought:To create an investment strategy, I need to analyze the historical stock data provided for each stock, predict their future trends, and calculate the optimal number of units to buy, the best time to buy them, and how long to hold them to maximize the expected profit within the given budget and risk tolerance.
>
> Action: I will first calculate the average price and the standard deviation (volatility) for each stock based on the historical data provided. This will help in understanding the general price movement and the risk associated with each stock.
>
> Action Input: Calculate the mean and standard deviation for each stock using the historical data provided.

---

> > ### Comment · Reviewer_b7iA · 2024-11-29
> >
> > Thank you for your detailed response. However, I still have several concerns:
> >
> > 1. **Methodology Separation**
> >    - While separating numerical computations from LLM reasoning seems intuitive, this strict division might be suboptimal
> >    - Recent research shows LLMs can perform meaningful numerical reasoning when properly prompted
> >    - The paper should justify why this particular separation is better than alternatives
> >
> > 2. **Generalizability Issues**
> >    - The claim that "scientists usually have limited questions" oversimplifies real-world complexity
> >    - The need to equip TS-Reasoner with domain-specific tools for each new application suggests limited scalability
> >    - A more systematic approach to domain adaptation should be proposed
> >
> > 3. **Baseline Comparison Fairness**
> >    - Using different models for baseline (ARIMA) and proposed method (TEMPO) raises fairness concerns
> >    - The superior performance might come from TEMPO's capabilities rather than the reasoning framework
> >    - A fair comparison should either use the same underlying model or justify why different models are appropriate
> >    - Moreover, it is worth noting that, apart from the Introduction and Related Work sections, I did not find specific descriptions in the paper on how to utilize the time series foundation model, which lacks detailed implements.
> >
> > 4. **ReAct Comparison**
> >    - The criticism of ReAct focuses on implementation details rather than fundamental methodological differences
> >    - The paper should explain why the proposed approach is structurally superior
> >    - Simply listing ReAct's limitations without proper analysis is insufficient
> >
> > 5. **Performance Stability**
> >    - The newly provided RMSE values show significant variations across different tasks
> >    - The large performance gaps (e.g., 3.82 vs 8.81 for Profit Percent) need explanation

---

> ### Author Response · Authors · 2024-11-29
>
> Dear Reviewer, thank you for your response!
>
> 1. Specialized numerical tools are designed to provide accurate computations where LLM excels at language understanding. TS-Reasoner leverages the strength of each and this separation facilitates the integration of domain-specific numerical tools, more specifically time series analysis tools in this case. To the best of our knowledge, recent works on numerical reasoning are focusing on mathematical/arithmetic computations. Unlike general numerical reasoning, time series analysis often involves capturing patterns such as trend, seasonality, temporal/lag dependency which are not exactly numerical reasoning. The use of pretrained time series foundation models—which process such characteristics internally—makes them a natural fit for integration within TS-Reasoner. Our reasoning is about decomposing tasks into actionable sub-modules and such an execution plan serves as a reasoning trace of how to approach the given question. This decomposition also supports error tracing. If you would like to suggest more applicable works on numerical reasoning with time series, we would be more than happy to check.
>
> 2. We acknowledge your concern that the statement "scientists usually have limited questions" may oversimplify real-world complexity. Our intent was to highlight the need for domain-specific tools tailored to distinct tasks. The nature of scientific inquiry varies significantly across domains, and the tools required often differ fundamentally. As such, a one-size-fits-all solution for domain adaptation may not be feasible. However, our modular design allows for flexibility in integrating domain-specific tools without requiring extensive reconfiguration. As mentioned previously, if the reviewer has specific question types or domain scenarios in mind, we welcome the opportunity to explore them and are open to extending our experiments accordingly.
>
> 3. Both TS-Reasoner and the baseline models are provided with the same input data and questions. The key distinction lies in how TS-Reasoner enhances reasoning capabilities through program decomposition and its integration with custom-wrapped time series foundation models. Such integration is a key advantage of TS-Reasoner as a hybrid approach which we will emphasize more clearly in the paper. Thank you for pointing out detailed descriptions of foundation model, we will make sure to add it to our final manuscript. We downloaded the pretrained TEMPO model and wrapped its inference code in a function inside the SinglePreOP module and hence is invoked by the SinglePreOP module with proper input arguments such as forecasting length and data.
>
> 4. ReAct relies on external databases and Wikipedia API and performs well on standard natural language QA and fact verification tasks. Such enhancement does not render useful in the context of processing time series data or performing time series analysis. Its design lacks the ability to process temporal dependencies or perform nuanced time series reasoning. TS-Reasoner’s modular approach, integrating domain-specific tools and pretrained time series models, directly addresses the unique challenges of time series reasoning. We ran the comparative analysis against ReAct and TS-Reasoner. We have also given an example from ReAct which illustrated the failure of ReAct to perform the multi-step time series reasoning task we proposed. We have also summarized the key limitation of ReAct and provided explanation for why it’s not suitable for our exciting new task. We kindly request clarification on what further proper analysis is requested, as we are eager to address your concerns fully.
>
> 5. We would like to clarify the perceived performance variations in RMSE. As mentioned in the previous response, RMSE values reflect differences in the underlying data, such as stock prices ranging from \$0 .25 in one task to \\$118 in another. These variations are inherent to the diverse questions and datasets used in our experiments. For each task, we randomly sample questions from our question generator with the specified constraint type (profit percent etc.). One question may contain data on a much smaller scale compared to data from another question. The time series data paired for questions in different tasks are not guaranteed to be the same. For each task, we do ensure the same set of questions was used across all model baselines to maintain a fair comparison.
>
> Again, we highly appreciate your engagement with us and hope this response addresses your concern.

---

> > ### Comment · Reviewer_b7iA · 2024-12-02
> >
> > Thank you for your comprehensive response. After carefully reviewing your reply, I believe this work is still in its preliminary stages and holds significant potential for further improvement. I look forward to seeing your future advancements.

---

### Official Review · Reviewer_o6zc · 2024-11-02

**Soundness:** 3
**Presentation:** 4
**Contribution:** 3
**Rating:** 6
**Confidence:** 2

**Summary:**

This paper proposes a new task, Compositional Time Series Reasoning (TSR), which focuses on handling complex, multi-step reasoning tasks on time series data. The paper introduces a novel approach called TS-Reasoner that utilizes LLMs to decompose complex tasks into structured programs and solve these tasks by calling predefined tool modules. Additionally, new datasets have been compiled for evaluation purposes.

**Strengths:**

**Novel Task Definition** The paper introduces a well-defined and relevant new task, TSR, which addresses a gap in the time series analysis literature by focusing on complex reasoning tasks.

**Interesting Approach** TS-Reasoner is an innovative approach that effectively combines LLMs with program-based decomposition, offering a promising solution for TSR. By leveraging the in-context learning and logical inference capabilities of LLMs while avoiding the direct management of numerical information, this method achieves impressive results.

**Comprehensive Evaluation** The paper presents a thorough evaluation of TS-Reasoner on datasets from two domains, demonstrating its effectiveness and superiority over CoT based methods.

**Weaknesses:**

**Limited Analysis of Error Cases** While the paper discusses error types, a more detailed analysis of specific error cases and their root causes would provide a deeper understanding of the method’s limitations.

**Dataset Size and Diversification** Although the paper presents evaluations on datasets from finance and energy domains, the size and diversity of these datasets are not clearly specified. It would be beneficial to provide more details about the dataset composition. Additionally, exploring the generalizability of TS-Reasoner to other domains would be valuable.

**Questions:**

See Weakness

**Details Of Ethics Concerns:**

No ethics concerns

---

> ### Author Response · Authors · 2024-11-21
>
> Dear Reviewer: Thank you for your thoughtful and detailed review. We appreciate the time and effort you invested in evaluating our work and providing constructive feedback. Below, we address your concerns and clarify key aspects of our approach.
>
> W1:Thank you for raising the point, we have performed a detailed error analysis and root cause trace on all error cases on tasks that did not achieve 100% success rate and collected specific error cases among all error cases for each task.
> Energy max, min load task: 100% of the error cases are due to constraint violation. The root cause is our refinement tool not good enough
> Energy variability task: 2/3 of all error cases are due to constraint violation where the root cause being the refinement tool not good enough. 1/3 of all error cases are due to the prediction result is of wrong length where the root case is the refinement module generated problematic code and incorrectly sliced the data.
> Stock risk, budget allocation task: 100% of all error cases are due to constraint violation: cost of investment exceeds budget where the root cause being the optimization module being not powerful enough. We also want to acknowledge that constrained multi dimension optimization is an extremely hard problem.
> Stock Profit task: 100% of all error cases are due to constraint violation. Within all error cases, 5% error cases are due to profit value not meeting the requirement and 95% of error cases are due to the cost of investment exceeding budget. Both root causes are the optimization module being not powerful enough.
> In summary, this analysis shed light on the future direction of further designing more powerful tools in applicable cases to boost TS-Reasoner's performance.
>
> W2: Thank you for raising the point! We have organized the dataset size as shown in table 8 (attached in the supplementary material pdf). The daily yahoo stock prices data involve stock prices for tickers from the earliest available data to September 2024. The hourly yahoo stock price data spans a week in September 2024 with 7 trading hours on each of the 5 business days in a week. The energy data contained electricity load from 6 major electricity grids in the United States (MISO, ERCOT,CAISO,NYISO,PJM, SPP) across 66 zones for 3 complete years (201 -2020) with a minute level frequency. The electricity load data is paired with corresponding weather variables. Additionally, we have taken steps to diversify the dataset by paraphrasing questions. In practical usage, users may ask the same question in various ways. To simulate this, we applied paraphrasing techniques using an LLM with diverse generation settings (e.g., temperature = 1, top_p = 1). This process helps increase the linguistic variety of the questions, making the dataset more robust and better suited for real-world applications.
>
> We sincerely thank you for the insightful comments and constructive feedback, which have helped us improve the clarity and depth of our work. We are happy to engage in further discussions.
>
> | **Dataset**           | **Number of CSVs** | **Avg Total Timestamps** | **Number of Variables** |
> |------------------------|---------------------|---------------------------|--------------------------|
> | **Daily Yahoo Stock**  | 6780               | 3785                      | 7                        |
> | **Hourly Yahoo Stock** | 5540               | 35                        | 7                        |
> | **Energy Data**        | 66                 | 872601                    | 11                       |
> | **Causal Data**        | 8                  | 529                       | 3--6                     |
>
> Table 8: Dataset Statistics of the constructed dataset. The exact number of time series are not
> calculated because it depends on randomly sampled sequence length when generating task instances.

---

> > ### Comment · Reviewer_o6zc · 2024-11-25
> > **Response to Rebuttal**
> >
> > Your responses have clarified key aspects and improved the presentation of your approach. I appreciate the effort you’ve made to address my comments, and I remain confident in the positive contributions of your work.

---

### Official Review · Reviewer_ro8i · 2024-11-04

**Soundness:** 2
**Presentation:** 4
**Contribution:** 2
**Rating:** 3
**Confidence:** 5

**Summary:**

Since many real-world tasks require additional reasonings in addition to time series forecasting, this paper introduces the task of compositional time series reasoning and designs a dataset for this task. The task includes three categories: 1) optimizing some objectives using the forecast results, 2) refining forecast results based on some given constraints, and 3) discovering the causal relationship among variables. The authors predefine a set of functions that a LLM can call, and directly prompts the LLM to solve the above tasks using these functions.

**Strengths:**

1. The paper is reasonably well-written. It is easy to follow and understand its approach and motivations.
2. The tasks proposed in the dataset are realistic and have practical values. The dataset will also be valuable to the community.

**Weaknesses:**

1. From my understanding, each type of question has a fixed template, where only the numbers indicated by "{}" will differ from question to question. Since this dataset might be valuable if future work decides to use it to finetune language models. I would suggest to ask off-the-shelf LLMs to rephrase the questions to make them more diverse. Currently, the contribution of this dataset is limited.
2. My biggest concern is that the dataset only proposes several types of questions. If there are only several types of questions, why would someone need an LLM to decide which programs to call? There should be some rules or some decision tree that can simply decide which programs should be called first. LLMs should be more beneficial if there are external language information in the task, or the task will benefit from some general world knowledge (e.g., a hot summer will increase ice cream sales). It is not clear from the current paper how applying LLMs will benefit this task. Please include additional case studies and explanations if so.

**Questions:**

1. Should the LLM have access to the list of the predefined tools and their explanations in the prompt (Section C.2)? If so, you might want to make this more clear in Section C.1.
2. The paper chooses to only prompt the LLM, instead of finetuning an LLM. I think teaching the model to read the time series directly should enable more power from LLMs. It would be interesting if you could show additional results on that.

Minor:
- Youtube link in the paper doesn't work. Please update it to the link in the repo.
- L348 on page 7, broken figure reference.
- L500 on page 10, "it addiditonally introduce ..."
- Some of the figure/table captions end with periods, and some don't. Please make them consistent.
- Table 3 is not captioned correctly: "Table 3: Overall performance of causal relationship recognition."
- L300 on page 6, in the question template, "for the future future length hours." It should be "{future length}". Same for L317, "variable names".

=================================================

Post Rebuttal:
Thanks to the authors for responding to my comments/questions. I will keep my score as 3.

The big picture this paper works on is quite interesting. This paper has the potential to be above the accept bar, but needs the following components that I do not think can be thoroughly addressed in the rebuttal:
1) Allow the reviewers access to your dataset so we can check the quality manually and see if there are any misunderstandings;
2) Introduce more operations and tasks that requires a true understanding of time series analysis tasks, instead of using LLMs merely as a tool to extract input parameters to a limited set of functions.
3) Finetune the LLM using a time series encoder instead of just tokenizing the time series values as text.

My main criticism on the current version is that there's only a very limited number of tasks, and each task has a very small solution space with only three to four possible operations. Also, I highly suspect the dataset itself is very redundant since finetuning doesn't show any improvement over simple prompting (that's why I want to see the entire dataset). I feel like the current version of this paper is too early-stage, and is more suitable for publishing at a workshop first.

---

> ### Author Response · Authors · 2024-11-21
>
> Dear Reviewer: Thank you for your thoughtful and detailed review. We appreciate the time and effort you invested in evaluating our work and providing constructive feedback. Below, we address your concerns and clarify key aspects of our approach.
>
> W1: Please see Q2
>
> W2: In its current version, TS-Reasoner is not limited to addressing a few specific questions; rather, we have already implemented support for tasks and scenarios such as time series numerical prediction, trend analysis, stability evaluation, decision-making, combinatorial optimization, and causal inference. These functionalities cover the mainstream tasks and scenarios in time series analysis with practical domain applications.
> Regarding the rule-based or decision-tree approaches you mentioned, their primary limitation is the inability to dynamically adapt to diverse time series tasks or automatically learn task-specific requirements. In contrast, TS-Reasoner can identify various time series tasks, perform complex reasoning and task decomposition, and dynamically parse critical information to generate function calls with appropriate arguments just by in-context learning samples.
> Additionally, rule-based systems struggle with flexibility when handling paraphrased or diverse question formulations. In contrast, TS-Reasoner leverages LLMs' inherent language capabilities to recognize similarities across differently expressed questions, ensuring robust and consistent performance. While incorporating external world knowledge into time series analysis remains inherently challenging, we are actively exploring this avenue as a future enhancement for TS-Reasoner.
>
> Q1:  In our current setup, we do not include detailed explanations of predefined tools in the prompt. Instead, we rely on in-context learning with question-program pairs that serve as examples. By providing similar questions and their corresponding programs as context, we observe that the task decomposer effectively learns to generate the appropriate program structure without explicit descriptions of the tools. This approach not only reduces prompt length but also demonstrates the LLM’s capability to learn from examples.
>
> Q2:  Thank you for raising this point! Upon continuous expansion efforts, we conducted additional experiments (attached in the supplementary material pdf table 6) involving fine-tuning using the LLAMA 3.1 8B Instruct model with question-program pairs. We utilized PEFT LoRA for efficient fine-tuning. Interestingly, our results show that the fine-tuned model performs comparably to ChatGPT 3.5 Turbo in a training-free in-context learning setting. We will revise our final manuscript to reflect this. Our choice of finetuning on question program pairs instead of on time series itself is supported by the current literature. There is a lack of evidence of superiority of finetuned LLM over developed time series foundation models in specific time series analysis task. As time series are numeric, we believe program-based approach is the best way to perform time series manipulation. LLM does not excel at handling numerical information and its performance in time series task is highly sensitive to how you tokenize the decimal point and numbers[1]. Additionally, finetuning LLM with time series data may lead to loss of natural language ability of LLM which we do not want to compromise as the main reason for leveraging LLM is its natural language capability.
>
> We shared the same thoughts with you on testing on diverse paraphrased questions as users in real world practical scenarios may ask the same question in different ways. We performed paraphrasing on testing datasets (shown in table 6) as users may ask the same questions in many different ways in practical situations. We fed original questions to the off-the-shelf LLAMA 3.1 8B Instruct model with the most diverse generation settings (temperature = 1, top_p = 1) to obtain paraphrased questions. The prompt we used for paraphrasing is:"Please rephrase the following question without changing its meaning: '{user_input}' Now, please paraphrase this quesiton without answering it. Put the paraphrased question in the following format: <begin paraphrase> ... <end paraphrase>." Our results remained consistent across all tasks compared to the current experimental setting with the exception of solely predicting stock price (which is the only one-step reasoning case). We discover that paraphrasing often mentions phrases like trend or volatility that sometimes lead to the task decomposer adding additional operations such as trend detection. The original intention of including this one-step test scenario was only to illustrate that baselines could perform similarly in one-step scenarios but fail with more reasoning steps. Our focus is still multi-step reasoning.
>
> [1]Gruver, Nate, Marc Finzi, Shikai Qiu, and Andrew G. Wilson. "Large language models are zero-shot time series forecasters." Advances in Neural Information Processing Systems 36 (2024).

---

> > ### Comment · Reviewer_ro8i · 2024-11-25
> >
> > Dear Authors, I have reviewed your responses to my questions, as well as the comments submitted by other reviewers. Thank you for your clarifications and additional experiments. However, I still believe that there are three main weaknesses. I see that my concerns are also shared by Reviewer n3Uv, Reviewer b7iA, Reviewer 3VA4, and Reviewer SXHV.
> >
> > 1. The dataset introduced is too simplistic and lacks diversity. Your new experiment on finetuning Llama shows that to perform well on your dataset, the model does not need to get access to the original time series. However, many real-world time series tasks definitely require treatments that are specific to the characteristics of the time series. Without seeing the actual time series, and not even including the explanations of the tools (based on your answer to Q1), the model can just do in-context learning --- this to me seems that the dataset is very likely to be overly simplistic and contains a lot of redundant samples.
> >
> > 2. You mentioned that TS-Reasoner is a first step towards LLM reasoning on time series. However, the dataset introduced lacks language diversity. If future works want to do finetuning to actually enhance the LLM's ability to read and perform ts analysis tasks, not having language diversity will make your dataset much less helpful.
> >
> > 3. The program-aided approach you highlighted as your main contribution also seems to lack novelty. It seems to be a straight-forward application of existing works on program-aided LLMs. The set of tools you defined for time series tasks are quite limited. It needs to be more comprehensive to be highlighted as the main contribution.
> >
> > I do not think the paper is ready for publication in its current form. However, the general idea is quite interesting. To improve the quality of the paper for your next submission, I suggest:
> > 1. When finetuning the LLM, use a time-series encoder, or even plotting the time series and feed into an image encoder would make a lot more sense than simply feeding the raw tokens into the LLM. When your dataset is diverse enough, the LLM definitely needs to see the time series in order to tell what sequence of tools to apply.
> >
> > 2. Your dataset must contain more open-ended tasks, rather than coming from four fixed tasks. Currently, since the tasks within each category are very similar, I still think that you can just define a simple decision tree for each category, and achieve very good performance. When your dataset involves more open-ended tasks, then the LLM needs to truly understand time series analysis to do well. Right now, it seems that the LLM is just memorizing patterns in a highly redundant dataset.
> >
> > 3. The dataset is not currently accessible. It is hard to check the quality and verify the diversity of the dataset.

---

> ### Author Response · Authors · 2024-11-21
>
> Minor: Thank you for pointing out our typo! We have fixed it in our revision and included the youtube link https://www.youtube.com/watch?v=Pjq1S4uT89w .
>
> We sincerely thank you for the insightful comments and constructive feedback, which have helped us improve the clarity and depth of our work. We are happy to engage in further discussions.
>
> |                                     |                     | **TS-Reasoner-C**         |            | **TS-Reasoner-L**       |            | **TS-Reasoner-L+ paraphrased data** |            |
> |-------------------------------------|---------------------|-----------|--------------------|-----------|--------------------|-----------|--------------------|
> | **Task**                            | **Reasoning Steps** | **SR(%)** | **MAPE(std)**      | **SR(%)** | **MAPE(std)**      | **SR(%)** | **MAPE(std)**      |
> | **Stock Future Price Prediction**  | 1                   | **100.0** | **0.042(0.030)**   | **100.0** | **0.042(0.030)**   | 20.00     | 0.046(0.030)       |
> | **Stock Future Volatility Prediction** | 2                | **100.0** | **0.748(0.691)**   | **100.0** | **0.748(0.691)**   | **100.0** | **0.748(0.691)**   |
> | **Energy Power w/ Max Load**       | 3                   | **97.87** | **0.101(0.339)**   | **97.87** | **0.101(0.339)**   | **97.87** | **0.101(0.339)**   |
> | **Energy Power w/ Min Load**       | 3                   | 97.83     | **0.084(0.104)**   | **100.00** | 0.086(0.103)      | **100.00** | 0.086(0.103)       |
> | **Load Ramp Rate in Energy Power** | 3                   | **100.0** | 0.060(0.153)       | 93.75     | 0.058(0.149)       | 97.91     | **0.053(0.144)**   |
> | **Load Variability Limit in Energy Power** | 3            | 93.88     | 0.288(0.385)       | **97.96** | **0.203(0.308)**   | 89.80     | 0.294(0.375)       |
>
> Table 6: The overall success rate and performance of TS-Reasoner variants. TS-Reasoner-C denotes TS-Reasoner with ChatGPT as task decomposer leveraging it's in context learning ability, TS-Reasoner-L denotes TS-Reasoner with finetuned LLAMA as task
> decomposer, TS-Reasoner-L  + paraphrased data denotesTS-Reasoner with LLAMA finetuned on para-
> phrased data as task decomposer evaluated on paraphrased data. SR stands for Success Rate; MAPE
> is the Mean Absolute Percentage Error. Bold indicates the best results

---

> ### Author Response · Authors · 2024-11-25
>
> Thank you very much for engaging with our feedback. We want to politely point out that there are possibly two major misunderstandings in your feedback:
>
>
> The first misunderstanding is about the dataset lacks language diversity. The questions we complied are the commonly asked important scientific questions in the relevant domains, which are extremely challenging even for domain experts and cannot be solved by existing general purpose LLMs. Language diversity is needed for general purpose LLMs, but there are only a limited number of scientific problems that the scientists and domain experts would work on. Our goal is to speed up the scientific discovery process by automating the complex inference tasks over time series data,  which usually takes domain experts weeks or months. For intermediate steps (such as forecasting, etc)of the multi-step analysis , existing time series foundation models could serve a good purpose.
>
> Also, our framework CAN be used to address the open-ended questions. We do not have a list of so-called open-ended questions  because except for the questions we collected, the rest are not the typically questions that the domain experts are concerned about (since otherwise we would include them already). It would be meaningless to generate arbitrary open-ended questions to simply serve the purpose of language diversity.
>
> We would also like to point out that human still have very limited understanding about time series. The process is still in iteration as we speak, which means we ask questions, then get good answers, improve our understanding and then we ask more questions. It is infeasible to list all questions before we finish the first iteration (which is the main purpose of this work).
>
> For your third concern, yes, we do plan to share the datasets.
>
> We hope that this clarify your concerns. Different from many submissions/papers which propose yet another transformer-based model for time series, our work aims to push the frontiers of time series research to the next level. When it is unexplored, the work will surely have weakness. Your reviews have helped improve the quality of our work, and we are grateful for your feedback and discussions.

---

> > ### Comment · Reviewer_ro8i · 2024-11-25
> >
> > Thanks for your quick response! I appreciate your effort to address my concerns.
> >
> > Based on Section C.1 in the appendix, the original TS-Reasoner does not have access to the raw time series values. Your finetuned TS-Reasoner does have access to these values, but it does not show any improvements over the original TS-Reasoner based on your previous comments.
> >
> > **For tasks that are actually time-consuming, the sequence of operations to apply should be dependent on the time series values and the unique characteristics of the given time series.** For example, for forecasting, we are given a large set of variables, but maybe the target variable we care about is only dependent on a few of these variables. For example, ice-cream sales should be dependent on the temperature, but not really dependent on the stock price. And maybe there are also some lag dependencies. The TS-Reasoner should be able to know which variables to look at, and maybe compute lag differences, etc, and those operations are all dependent on the time series values.
> >
> > However, for each of your four tasks in Table 3, there are a limited number of possible operations. For example, causal discovery only has two operations: GrangerCausalMatrixOP, GetCausalRelationsOP. Energy forecasting only has three operations: MultiPreOP, ApplyOP, RefGenOP. To solve these tasks, it seems to me that most of the time we will apply these operations in a fixed order, and the differences is only the input parameters to these functions (for example, how many time steps we want to forecast). If I am wrong, please provide some statistics on how many different sequences of operations are used, and what is the frequency of calling each sequence of operations.
> >
> > Overall, it does not seem like given these same set of tools, the complexity of these tasks will require domain experts weeks or months to work on. To do causal discovery and the two functions defined, we always want to first apply GrangerCausalMatrixOP and then GetCausalRelationsOP. The same holds for forecasting tasks.
> >
> > I still think that your topic is interesting and can potentially be valuable. However, I suggest to 1) allow the reviewers access to your dataset so we can check the quality manually and see if there are any misunderstandings, 2) introduce more operations and tasks that requires a true understanding of time series analysis tasks, instead of using LLMs merely as a tool to extract input parameters to a limited set of functions, 3) finetune the LLM using a time series encoder instead of just tokenizing the time series values as text.

---

> ### Author Response · Authors · 2024-11-29
>
> Dear reviewer, we sincerely thank you for the suggestions. We may have introduced more confusions with previous experiments. We would like to clarify it. The aforementioned two versions are variants of TS-Reasoner with different task decomposer backbone either leveraging ChatGPT’s in context learning ability or Finetuned Llama for task decomposition. For standalone LLM like ChatGPT 4 turbo, the performance is considerably inferior (as shown in table 2 and also the table below). For directly finetuning together with raw time series for end-to-end task execution, we performed this experiment on Llama 8b Instruct and it’s much worse than TS-Reasoner due to the output space complexity of different task instances as as shown in table below. Neither standalone LLM and finetuned LLM for end task execution are sufficient for multi step time series reasoning.
>
> |                                     |                     | **TS-Reasoner**         |            | **Finetuned Llama**       |            | **ChatGPT** |            |
> |-------------------------------------|---------------------|-----------|--------------------|-----------|--------------------|-----------|--------------------|
> | **Task**                            | **Reasoning Steps** | **SR(%)** | **MAPE(std)**      | **SR(%)** | **MAPE(std)**      | **SR(%)** | **MAPE(std)**      |
> | Stock Future Price Prediction         | 1                   | **100.0**               | 0.042 (0.030)              | 4.00                     | **0.038 (0.011)**           | **100.0**         | 0.058 (0.047)         |
> | Stock Future Volatility Prediction    | 2                   | **100.0**               | **0.748 (0.691)**          | 0                        | -                            | 88.00             | 0.865 (0.158)         |
> | Energy Power w/ Max Load              | 3                   | **97.87**               | **0.101 (0.339)**          | 19.15                    | 0.149 (0.312)               | 53.20             | 0.120 (0.230)         |
> | Energy Power w/ Min Load              | 3                   | **97.83**               | **0.084 (0.104)**          | 15.22                    | 0.317 (0.437)               | 54.30             | 0.279 (0.500)         |
> | Load Ramp Rate in Energy Power        | 3                   | **100.0**               | **0.060 (0.153)**          | 31.25                    | 0.125 (0.191)               | 75.00             | 0.062 (0.170)         |
> | Load Variability Limit in Energy Power| 3                   | **93.88**               | 0.288 (0.385)              | 16.32                    | 0.821 (1.212)               | 55.10             | **0.243 (0.391)**     |
>
> **Table 11:** The overall success rate and performance of TS-Reasoner compared to Llama and ChatGPT. TS-Reasoner-C denotes TS-Reasoner with ChatGPT as task decomposer leveraging its in-context learning ability. Finetuned Llama denotes Llama fine-tuned on the question and input time series, and outputs the final answer for end task completion. ChatGPT denotes directly prompting ChatGPT with CoT for end-task completion. SR stands for Success Rate; MAPE is the Mean Absolute Percentage Error. Bold indicates the best results.

---

> ### Author Response · Authors · 2024-11-29
>
> In terms of a limited set of tools, our forecasting task leveraged TEMPO model, selected for its strong zero shot capability. There are no hyperparameter selection since it’s a pretrained model for zero shot inference. With lag information, many time series models have incorporated it internally and can be easily included into the toolbox. With the development of general purpose time series models, time series foundation models are generally applicable to imputation, classification, anomaly detection tasks [1][2]. In our example illustrations, we were mainly using time series forecasting foundation models as there is much more development in this field [3][4][5]. Your point on developing more open-ended tasks is well-taken and it’s part of our future work and continuous expansion effort of TS-Reasoner.We have also revised the introduction and conclusion accordingly to better position the work. At the current stage, TS-Reasoner is already one-step forward and opening up the possibility for more sophisticated approaches for tackling real world multi-step time series reasoning. This is a new problem in time series that has not been well studied or studied at all. TS-Reasoner provides a new way of thinking about the problem of compositional time series reasoning. We are not overclaiming the capability of TS-Reasoner and fully acknowledge the limitation. All your suggestions help us improve.
>
> [1]Zhou, Tian, Peisong Niu, Liang Sun, and Rong Jin. "One fits all: Power general time series analysis by pretrained lm." Advances in neural information processing systems 36 (2023): 43322-43355.
> [2] Gao, Shanghua, Teddy Koker, Owen Queen, Thomas Hartvigsen, Theodoros Tsiligkaridis, and Marinka Zitnik. "UniTS: A unified multi-task time series model." In The Thirty-eighth Annual Conference on Neural Information Processing Systems. 2024.
> [3] Garza, Azul, and Max Mergenthaler-Canseco. "TimeGPT-1." arXiv preprint arXiv:2310.03589 (2023).
> [4]Cao, Defu, Furong Jia, Sercan O. Arik, Tomas Pfister, Yixiang Zheng, Wen Ye, and Yan Liu. "Tempo: Prompt-based generative pre-trained transformer for time series forecasting." arXiv preprint arXiv:2310.04948 (2023).
> [5]Ansari, Abdul Fatir, Lorenzo Stella, Caner Turkmen, Xiyuan Zhang, Pedro Mercado, Huibin Shen, Oleksandr Shchur et al. "Chronos: Learning the language of time series." arXiv preprint arXiv:2403.07815 (2024).
>
>
> Here are a few sample question in plain text format as no links or attachments are allowed and it contains time series paired with question instance and ground truth data. We are open to new questions if you are willing to suggest questions you have in mind. We are open to running experiments that the reviewer suggests. The data sources for time series data itself are public (yahoo finance api for stock data and paper “PSML: A Multi-scale Time-series Dataset for Machine Learning in Decarbonized Energy Grids” for energy data).

---

### Official Review · Reviewer_n3Uv · 2024-11-04

**Soundness:** 2
**Presentation:** 3
**Contribution:** 3
**Rating:** 6
**Confidence:** 4

**Summary:**

This paper studies a new task of compositional time series reasoning which involves conducting multistep reasoning from time series data. The proposed approach leverages LLM to decompose a complex task into a series of programs that call specific time-series models, numerical methods or custom generation modules. The experiments conducted across three types of tasks—decision-making, compositional question answering, and causal mining—demonstrate promising results when compared to directly prompting LLMs.

**Strengths:**

1. The paper studies a new task of time-series reasoning, constructs specific scenarios that cover decision-making, compositional question answering, causal mining, and composes new datasets in finance and energy domains.

2. The paper proposes an effective method that leverages LLMs for task decomposition, calling dedicated time-series models, numerical methods or custom generation modules. Experiments on multiple tasks show clear improvement over direct LLM prompting.

**Weaknesses:**

1. It is unclear how scalable the proposed approach is to new tasks and scenarios. The model might not be able to generalize to new tasks without manually-designed prompts for in-context learning. Additionally, the modules studied in the paper are limited and do not cover real-world scenarios, and the errors might increase as the number of modules increases.

2. The compositional question answering on financial time series appears similar to a traditional forecasting scenario. How does the model compare with existing state-of-the-art forecasting model for this task?

3. How to trace errors back to specific steps within the reasoning chain?

4. Are LLM-generated synthetic data for causal mining reliable?

3. In Line 348, there is a typo of "as illustrated in Fig".

**Questions:**

How does TS-Reasoner compare to o1?

---

> ### Author Response · Authors · 2024-11-21
>
> Dear Reviewer: Thank you for your thoughtful and detailed review. We appreciate the time and effort you invested in evaluating our work and providing constructive feedback. Below, we address your concerns and clarify key aspects of our approach.
>
> W1: In our current work, we acknowledge the use of predefined prompts for in-context learning; however, this design choice is intended to establish a clear initial framework. The use cases were carefully curated based on surveying important questions that domain experts ask in these domains [1][2][3]. This approach reflects our intention to develop a practical tool that scientists and engineers can use for domain-specific tasks and achieve good results, rather than building a general-purpose model that may be able to handle all tasks but perform poorly on each. On the note of generalizability, we emphasize that our **program-based reasoning approach inherently supports modular expansion**. This allows new tasks to be accommodated by adding relevant modules without redesigning the entire system. Moreover, our experiments primarily focus on real-world scenarios from energy and financial domains, with the exception of causal reasoning, which uses well-grounded synthetic data.
>
> W2: TS-Reasoner directly leverages state-of-the-art (SOTA) forecasting models as foundational components. Upon continuous expansion of the toolbox, foundation models such as chronos, timegpt, lagllama, tempo, are all included in our toolbox. However, the key distinction lies in the integration of these forecasting modules with additional multi-step reasoning tasks, such as constraint handling and optimization, which cannot be addressed by standalone forecasting models and are addressed poorly by standalone general purpose LLM. This holistic approach enables our framework to handle more sophisticated tasks that go beyond forecasting.
>
> W3: We agree that error traceability is crucial for understanding model behavior. Our program-based reasoning framework facilitates error tracing by maintaining detailed logs for each module execution, capturing both the intermediate outputs and the sequence of program steps. This explicit logging enables us to identify issues at specific modules (e.g., forecasting, optimization), rather than treating the process as a black box. For instance, errors during parsing variables or module execution are captured before reaching the evaluation protocol, and a specific error statement is returned. The evaluation protocol also provides detailed error messages, such as length mismatches or format inconsistencies, allowing us to diagnose failures effectively.
>
> W4: we highlight that our synthetic data generation is guided by well-established domain principles and statistical relationships, ensuring that the generated datasets reflect plausible real-world dynamics and are grounded in real world scenarios.
> ### Real-World Background for Variables:
> 1. **Marketing Spend (A)**: Budget allocated to marketing activities, influencing brand awareness and product demand.
> 2. **Product Development (B)**: Investment in product development, influencing the quality and variety of products, which impacts sales.
> 3. **Market Trends (C)**: General trends in the market, such as consumer preferences or technological advancements.
> 4. **Operational Costs (D)**: Costs associated with the operations of a company, such as manufacturing and logistics.
> 5. **Competitor Activity (E)**: Activities by competitors, such as new product launches or pricing strategies.
> 6. **Sales (F)**: Total sales revenue generated by the company.
> ### Relation Matrix Interpretation:
> - A influences C.
> - B influences A, D, and F.
> - C influences E.
> - D influences only itself.
> - E influences C and D.
> - F influences B.
>
> The example output illustrates its ability to generate realistic data by capturing the interactions between key variables in a business scenario. For instance, Marketing Spend (A) influences Market Trends (C), while Product Development (B) impacts Marketing Spend (A), Operational Costs (D), and Sales (F), highlighting the interconnected nature of investment, cost, and revenue. Furthermore, Market Trends (C) affect Competitor Activity (E), which in turn influences both Market Trends (C) and Operational Costs (D), showcasing the dynamic feedback between market shifts and competition. Additionally, Sales (F) provide critical feedback to Product Development (B), ensuring that sales performance drives future innovation. This causal chain demonstrates the model’s capability to replicate complex real-world relationships in data generation.

---

> ### Author Response · Authors · 2024-11-21
>
> [1] D’Amico, Guglielmo, Filippo Petroni, and Salvatore Vergine. "Ramp rate limitation of wind power: An overview." Energies 15, no. 16 (2022): 5850.
>
> [2] Zidan, Aboelsood, and Ehab F. El-Saadany. "Distribution system reconfiguration for energy loss reduction considering the variability of load and local renewable generation." Energy 59 (2013): 698-707.
>
> [3] Lee, Tae Kyun, Joon Hyung Cho, Deuk Sin Kwon, and So Young Sohn. "Global stock market investment strategies based on financial network indicators using machine learning techniques." Expert Systems with Applications 117 (2019): 228-242.
>
> Q1: We conducted a series of comparative experiments between our TS-Reasoner and OpenAI's O1-Preview across three key tasks: decision-making, compositional QA, and causal relationship recognition. We have attached the results in the supplementary material pdf.
> As shown in Table 4, for decision-making tasks, our method outperforms o1 in both Profit Percent and Budget Allocation scenarios, demonstrating its superior ability to handle complex optimization under varying constraints. In managing risk tolerance case, TS-Reasoner continues to outperform o1 on success rate indicating satisfactory solutions are consistently obtained (less constraint violation). We do notice that o1 has a higher average profit among the cases that it succeeded indicating a more risking behavior (failing more cases but earning high profits among the few cases succeeded). It is worth emphasizing that our method excels in decision-making scenarios requiring stable, balanced reasoning, showcasing its robustness and adaptability.
> In compositional QA tasks, the comparison results are presented in Table 5. For Stock Future Price Prediction and Volatility Prediction Task, TS-Reasoner consistently outperforms o1, demonstrating the advantage of integrating time series specific foundation models. For more complex reasoning tasks involving constraints, TS-Reasoner demonstrated a much stronger ability to adhere to domain knowledge and constraint where the success rate surpasses o1 by a large margin.
> Finally, in the causal relationship recognition task, as shown in Table 7, TS-Reasoner demonstrates clear superiority over o1 in both success rate and accuracy. This indicates that our method is particularly well-suited for identifying and reasoning about causal dependencies, a crucial capability in many real-world applications.
> Overall, even when compared with the highly advanced reasoning capabilities of o1, TS-Reasoner demonstrated superiority over o1 across tasks, especially in tasks that require incorporating constraints/domain knowledge. These results underline the strength and versatility of TS-Reasoner, further validating its effectiveness in tackling a wide range of reasoning-driven tasks in domains with important time series analysis applications. We acknowledge that o1 is a very strong model with reasoning capabilities and is very generalizable. This comparative experiment also echoes with our building philosophy of creating TS-Reasoner as a practical tool for domain experts in practical scenarios rather than a general purpose model that could handle all tasks but perform poorly on domain specific tasks.
>
> Minor: Thank you for pointing out our typo! We have fixed it in our revision.
>
> We sincerely thank you for the insightful comments and constructive feedback, which have helped us improve the clarity and depth of our work. We are happy to engage in further discussions. We are attaching the markdown version of the table for your convenience.
>
>
> |                       | **TS-Reasoner**     |           |           | **o1-preview**    |           |            | **ReAct**   |           |             |
> |-----------------------|-----------|-----------|-----------|-----------|-----------|------------|-----------|-----------|-------------|
> | **Task Requirement** | **SR(%)** | **AAP**   | **RAP**   | **SR(%)** | **AAP**   | **RAP**    | **SR(%)** | **AAP**   | **RAP**     |
> | **Profit Percent**    | **59.2**  | **243.31**| **32.34** | 6.1       | 12.53     | -198.43    | 0.0       | -         | -           |
> | **Risk Tolerance**    | **96.0**  | 54.54     | -46.04    | 18.0      | **124.72**| **24.14**  | 4.0       | -0.04     | -100.63     |
> | **Budget Allocation** | **90.0**  | **37.12** | **7.57**  | 28.0      | -195.96   | -225.50    | 4.0       | 15.70     | -13.84      |
>
> Table 4: The success rate and performance of TS-Reasoner against additional baselines on desicion
> making. SR stands for Success Rate; AAP stands for Absolute Average Profit. RAP is the Relative
> Average Profit compared to vanilla strategy. In Profit Percent and Budget Allocation task, we aim at
> improving the profit. Thus positive RAP is expected. In Risk Tolerance, the model is required to first
> ensure the risk and minimize the profit reduction. A negative RAP indicates a more conservative
> model in terms of risk management compared to vanilla strategy. Bold indicates the best results.

---

> ### Author Response · Authors · 2024-11-24
>
> We are attaching the markdown version of the new tables for your convenience.
>
> |                                     |                     | **TS-Reasoner**      |            | **o1-preview**      |            | **ReAct**          |                    |
> |-------------------------------------|---------------------|-----------|--------------------|-----------|--------------------|-----------|--------------------|
> | **Task**                            | **Reasoning Steps** | **SR(%)** | **MAPE(std)**      | **SR(%)** | **MAPE(std)**      | **SR(%)** | **MAPE(std)**    |
> | **Stock Future Price Prediction**  | 1                   | **100.0** | **0.042(0.030)**   | 100.0     | 0.053(0.031)       | 48.00     | 0.043(0.023)       |
> | **Stock Future Volatility Prediction** | 2                | **100.0** | **0.748(0.691)**   | 100.0     | 0.750(0.533)       | 46.00     | 1.123(0.882)       |
> | **Energy Power w/ Max Load**       | 3                   | **97.87** | 0.101(0.339)       | 78.72     | **0.095(0.198)**   | 21.28     | 0.136(0.292)       |
> | **Energy Power w/ Min Load**       | 3                   | **97.83** | **0.084(0.104)**   | 76.09     | 0.218(0.352)       | 36.96     | 0.374(0.796)       |
> | **Load Ramp Rate in Energy Power** | 3                   | **100.0** | **0.060(0.153)**   | 91.67     | 0.076(0.179)       | 29.17     | 0.131(0.273)       |
> | **Load Variability Limit in Energy Power** | 3            | **93.88** | 0.288(0.385)       | 89.80     | **0.169(0.290)**   | 26.53     | 0.268(0.360)       |
>
> Table 5: The overall success rate and performance of our model against additional baselines on
> compositional QA. SR stands for Success Rate; MAPE is the Mean Absolute Percentage Error.
> Bold indicates the best results.
> |                         | **TS-Reasoner**     |            |            | **o1-preview**     |            |            | **ReAct**   |            |            |
> |-------------------------|-----------|------------|------------|-----------|------------|------------|-----------|------------|------------|
> | **Task Requirement**   | **SR(%)** | **ACC(%)** | **SSR(%)** | **SR(%)** | **ACC(%)** | **SSR(%)** | **SR(%)** | **ACC(%)** | **SSR(%)** |
> | **Causal Relationship** | **100.0** | **79.15**   | **8.0**    | 82.0      | 74.08      | 4.0        | 50.0      | 76.13      | 0.0        |
>
> Table 7: The success rate and performance of TS-Reasoner against other baselines on causal relationship recognition. SR stands for Success Rate; ACC stands for Accuracy; SSR stands for Strict
> Success Rate. Bold indicates the best results.

---

> > ### Comment · Reviewer_n3Uv · 2024-11-27
> >
> > Thank the authors for the detailed response and the additional experiments. I believe this work has positive contributions and will keep my current score.

---

### Note · Authors · 2025-01-22

I have read and agree with the venue's withdrawal policy on behalf of myself and my co-authors.